# Global burden and risk factors of peptic ulcer disease between 1990 and 2021: An analysis from the global burden of disease study 2021

Wende Hao[1], Chaoyue Zheng[2], Zhenjun Wang[3], Huachong Ma [1]*

1 Department of Emergency Abdominal Surgery, Beijing Chaoyang Hospital, Capital Medical University, Beijing, China, 2 Department of Urology, Beijing Chaoyang Hospital, Capital Medical University, Beijing, China, 3 Department of General Surgery, Beijing Chaoyang Hospital, Capital Medical University, Beijing, China

* mhchong0660@ccmu.edu.cn

## Abstract

### Background

Peptic ulcer disease (PUD) is a chronic gastrointestinal disorder that may present acutely due to complications and poses significant clinical and economic challenges. Understanding the global burden of PUD and its contributing risk factors is essential for developing targeted prevention strategies. Therefore, our research aimed to comprehensively evaluate the epidemiological characteristics and associated risk factors of PUD, thereby providing evidence to support policymakers in formulating appropriate health policies.

### Methods

The data on PUD were retrieved from the Global Burden of Diseases, Injuries, and Risk Factors Study (GBD) 2021. Incidence, prevalence, deaths and disability-adjusted life years (DALYs) were metrics used to measure PUD burden. The population attributable fractions (PAFs) were used to calculate the percentage contributions of primary potential risk factors to PUD deaths and DALYs.

### Results

The global incidence and prevalence cases of PUD increased by 11.1% and 8.8%, respectively, between 1990 and 2021. In contrast, the number of deaths and DALYs decreased by 15.94% and 27.8%, respectively, during the same period. The global age-standardized rates (ASRs) for incidence, prevalence, deaths and DALYs associated with PUD decreased by 40.3%, 41.1%, 61.5%, and 63.1%, respectively, between 1990 and 2021. Men exhibited higher numbers and ASRs of incidence, prevalence, deaths, and DALYs associated with PUD than women across most age cohorts in 2021.The average annual percentage change (AAPC) in age-standardized

**Data availability statement:** The analyzed datasets generated during the study are available from the Global Burden of Disease, Injuries and Risk Factors Study (http://vizhub.healthdata.org/gbd-results/).

**Funding:** This study was funded by the Wu Jieping Medical Foundation Special Fund for Clinical Research in the form of a grant [320.6750.2022-07-15] to HM.

**Competing interests:** The authors declare no competing interests.

incidence (ASIR), prevalence (ASPR), deaths (ASMR), and DALYs (ASDR) rates for PUD were −1.65 (95% confidence interval (CI): −1.69, −1.61), −1.69 (95% CI: −1.74, −1.63), −3.02 (95% CI: −3.13, −2.91) and −3.17 (95% CI: −3.24,-3.10), respectively, from 1990 to 2021 on a global scale. In 2021, negative associations were observed globally among the ASIR, ASPR, ASMR, ASDR and the Socio-Demographic Index (SDI). Based on the ARIMA model, we projected that the global ASIR, ASPR, ASMR, and ASDR for PUD will exhibit decreasing trends from 2022 to 2040 for both sexes. We also identified smoking as the primary risk factor associated with PUD-related DALYs and deaths in both sexes in 1990 and 2021.

## Conclusion

Significant advancements have been noted in reducing the global burden of PUD. Nonetheless, significant geographical and gender disparities exist in PUD numbers and ASRs, suggesting that a substantial portion of the population still lacks access to quality healthcare or experiences variations in risk factors for PUD. Thus, precise prevention strategies are essential to mitigate the disease burden of PUD.

---

## Introduction

Peptic ulcer disease (PUD) refers to an acid-related injury of the digestive tract, characterized by a mucosal break that extends to the submucosa [1–3]. Peptic ulcers are typically found in the stomach or proximal duodenum, but they may also occur in the esophagus or Meckel's diverticulum [4,5]. PUD exhibits a lifetime prevalence of approximately 10% within the general population, accompanied by an annual incidence rate ranging from 0.1% to 0.2% [6]. Given the nonspecific symptoms of PUD, its assessment and treatment require clinical caution to prevent severe complications, such as bleeding, perforation and gastrointestinal obstruction, all of which may necessitate urgent endoscopic or surgical intervention [7–10].

Several epidemiological studies have demonstrated a significant decreasing trend in the incidence, rates of hospital admission, and mortality associated with PUD over the past 20–30 years due to the application of new anti-PUD therapies [11–13]. However, the extensive use of nonsteroidal anti-inflammatory drugs (NSAIDs) and aspirin, combined with smoking and alcohol consumption have been recognized as a set of risk factors and have significantly reshaped the landscape of PUD in recent years [14,15] Meanwhile, recent researches indicated that regions or countries with lower socioeconomic status may bear a disproportionately high burden of PUD based on results from GBD 2019 [16,17].

The GBD is a comprehensive global epidemiological investigation that assesses and monitors the health status of populations worldwide, which offers summary data for researchers and public health organizations to facilitate both policy-making and scientific research [18–20]. Since the update of the GBD 2021 database, there has been no comprehensive report on the epidemiology and trends of PUD. In this study, we analyzed PUD burdens in 204 countries or territories from 1990 to 2021 based

on data from the GBD 2021, and also predicted the PUD burdens until 2040 globally. Additionally, we provided a detailed insight into the risk factors contributing to the burden of PUD by sex wordwide. Through the comprehensive evaluation of global landscape of PUD, we hope to establish a conceptual framework for a deeper understanding of PUD's impact on population health and guide the development of global preventive strategies and the allocation of healthcare resources.

## Materials and methods

### Overview and data acquisition of GBD

The GBD study was developed to deliver comprehensive and comparable global health estimates concerning causes of death, injuries and risk factors [21]. Meanwhile, the GBD 2021 study estimated 369 causes of death and injuries, along with 87 risk factors at the global level, by region, and for 204 countries and territories [22]. We utilized data from the GBD 2021 to estimate the trends in the disease burden of PUD using four standard epidemiological measures: prevalence, incidence, deaths, and DALYs across specific groups of locations by sex, age, and year. The numbers and ASRs were analyzed to compare PUD burden trends across different cohorts. Graphs were also generated to illustrate the distribution and trends of change in global, regional, and national disease burdens attributable to PUD.

### Definition of SDI and uncertainty

SDI was utilized in this research to examine the relationship between a country's socioeconomic development status and the ASRs of PUD. The SDI ranges from 0 (indicating low development) to 1 (indicating high development). Alternatively, it can be categorized into five quintiles: low, low-middle, middle, high-middle, and high, based on their respective scores [23]. The uncertainty interval (UI) analysis was employed to account for the potential heterogeneity arising from both sampling error and non-sampling variance. The 95% UIs were calculated by drawing 1000 samples from the posterior distribution at each step of the modeling process and were reported as the 2.5th and 97.5th percentiles for each estimate [24].

### Joinpoint regression analysis

The joinpoint regression model comprises a series of linear statistical models utilized to assess the trends in disease burdens attributable to PUD over time.

This model estimates changes in illness rates using the least square method, which avoids the subjectivity of traditional trend analyses based on linear trends. The turning point of the shifting trend is identified by calculating the squared sum of the residual errors between the estimated and actual values. We also calculated the annual percentage change (APC) and AAPC, and assessed the statistical significance of trend fluctuations by comparing the APC or AAPC to zero. A P-value of less than 0.05 was considered statistically significant.

### Trend prediction

The autoregressive integrated moving average (ARIMA) model is composed of the autoregressive model and the moving average model [25]. The underlying assumption is that the data series are time-dependent random variables, with autocorrelation characterized by the ARIMA model, allowing future values to be predicted based on past values [26]. We employed the ARIMA model to forecast the ASR of PUD for both sexes from 2022 to 2040.

### Risk factor estimation

The Population Attributable Fraction (PAF) represents the proportion of an outcome that would be eliminated if a risk factor were reduced to the theoretical minimum risk exposure level [27]. The GBD 2021 utilized the comparative risk assessment framework to estimate DALYs and deaths for PUD attributable to specific risk factors. The detailed methods of estimation are described elsewhere [28,29].

## Ethical considerations

The study was based on publicly available data from the GBD 2021 and did not involve access to any individual-level data. As such, ethical approval was not required.

## Statistical analysis

All calculations and figures were conducted using EXCEL 2021 (Microsoft Corporation) and R software (version 4.3.2). $P<0.05$ was considered statistically significant.

## Results

### Global and regional burden of PUD

Globally, as shown in Tables 1 and 2, the incidence and prevalence cases of PUD in 2021 were 2,854,370 (95% UI: 2,438,231–3,264,252) per 100,000 and 6,567,782 (95% UI: 5,798,379-7,597,596) per 100,000, respectively, reflecting increases of 11.05% and 8.77% since 1990. However, the ASIR and the ASPR both decreased from 57.14 (95% UI: 48.61 to 66.73) per 100,000 and 132.97 (95% UI: 116.22 to 154.17) per 100,000 in 1990 to 34.1 (95% UI: 29.12 to 38.97) per 100,000 and 78.27 (95% UI: 69.02 to 90.75) per 100,000 in 2021. Between 1990 and 2021, the number of deaths attributed to PUD decreased from 273,872 (95% UI: 247,312–299,718) per 100,000–230,217 (95% UI: 193,005–270,858) per 100,000, representing a 15.94% reduction in global PUD-related deaths. Meanwhile, the global ASMR of PUD also showed a decreasing trend, at 7.14 (95% UI:6.41–7.82) per 100,000 in 1990 and 2.75 (95% UI:2.31–3.24) per 100,000 in 2021. Similar trends were also observed in DALYs cases and the ASDR. Among GBD regional areas, Oceania had the highest ASIR(65.40 (95% UI: 57.19–74.46) per 100,000) and ASPR(152.22 (95% UI: 133.42–174.34) per 100,000) in 2021. Besides, the highest ASMR and ASDR of PUD occurred in Central Sub-Saharan Africa (7.17 (95% UI: 4.24 to 10.15) per 100,000) and Eastern Sub-Saharan Africa (198.29 (95% UI:121.89 to 265.39) per 100,000) in 2021, respectively.

Age-specific numbers and rates of incidence, prevalence, deaths and DALYs for PUD by gender in 2021 were shown in Fig 1. The numbers of incidence and prevalence for PUD both peaked between the ages of 55–59 in males and 65–69 in females. Additionally, males had higher values than females between the ages of 20–74. PUD-related deaths peaked at ages 70–74 in males and 75–79 in females. Males had more deaths than females across age groups above 20–24, except in those aged 95 and older. The DALYs cases of PUD peaked between the ages of 55–59 in males and 65–69 in females, and males had more DALYs than females between the ages of 25–79. The age-specific incidence rate of PUD increased with age in both sexes, except for females aged 15–34. The age-specific prevalence rate of PUD increased with age in both sexes up to 79 years, except for females aged 20–34. The age-specific mortality rate began to increase after age 34 in males and age 44 in females, respectively. The age-specific DALY rate for males began to increase after age 14, whereas in females, it generally rose with age, except between ages 20–34.

From 1990 to 2021, both the numbers and ASRs of incidence, prevalence, deaths and DALYs for PUD were consistently higher in males than in females. However, the difference between the two groups diminished over time, primarily because the numbers and ASRs in males declined at a faster rate than those in females (Fig 2). Joinpoint regression analyses of the segmental trends in ASRs for PUD wordwide from 1990 to 2021 were also shown in Fig 3. The ASIR significantly decreased over time in both males (1990–1995 APC=−1.56, 1995–2006 APC= −1.79, 2006–2011 APC = −2.56, 2011–2014 APC=−2.96, 2014–2018 APC=−1.82, 2018–2021 APC=−0.19) and females (1990–1994 APC=−1.53, 1994–2001 APC= −1.01, 2001–2006 APC=−1.46, 2006–2014 APC= −2.30, 2014–2018 APC = −1.26) (Figs 3A and 3E). The ASPR also exhibited a significant decline over time for both males (1990–1996 APC=−1.67, 1996–2006 APC= −1.83, 2006–2011 APC = −2.57, 2011–2014 APC=−3.05, 2014–2018 APC=−1.77, 2018–2021 APC=−0.16) and females (1990–1994 APC=−1.56, 1994–2001 APC= −1.03, 2001–2006 APC=−1.56, 2006–2014 APC= −2.32, 2014–2018

Table 1. Numbers and age-standardized incidence, prevalence, deaths and DALYs rates of peptic ulcer disease in 1990.

| Characteristics | Incident case no. (95%UI) | ASIR/100,000 (95% UI) | Prevalence case no. (95%UI) | ASPR/100,000 (95% UI) | Deaths case no. (95%UI) | ASMR/100,000 (95% UI) | DALYs case no. (95%UI) | ASDR/100,000 (95% UI) |
|---|---|---|---|---|---|---|---|---|
| Global | 25704413 (2161831,2997880) | 57.14 (48.61,66.73) | 6038112 (5268704,7064284) | 132.97 (116.22,154.17) | 273872 (247312,299718) | 7.14 (6.41,7.82) | 8394780 (7603519,9210479) | 193.82 (175.79,212.14) |
| **SDI** | | | | | | | | |
| High SDI | 443846 (378354,518641) | 42.22 (36.29,48.9) | 1047697 (909514,1218869) | 99.39 (86.54,115.15) | 37224 (34332,38752) | 3.39 (3.11,3.53) | 767841 (731642,799274) | 71.22 (67.88,74.16) |
| High-middle SDI | 465252 (390682,555691) | 44.53 (37.65,52.69) | 1080318 (929968,1273589) | 102.57 (88.22,120.59) | 39083 (35470,43166) | 4.23 (3.82,4.67) | 1075287 (981505,1179438) | 106.72 (97.62,117.17) |
| Middle SDI | 796528 (658968,940458) | 61.88 (52.23,73.53) | 1875446 (1622282,2211919) | 143.61 (123.39,169.79) | 72196 (63912,81485) | 7.85 (6.88,8.82) | 2197829 (1952833,2455155) | 190.52 (169.12,213.21) |
| Low-middle SDI | 637902 (525648,743070) | 77.81 (65.8,90.35) | 1510337 (1307375,1773592) | 178.29 (157.16,207.12) | 87752 (73171,105625) | 14.21 (11.69,17.37) | 3007201 (2571474,3532661) | 400.31 (338.45,474.29) |
| Low SDI | 224839 (184265,263050) | 65.49 (55.39,75.58) | 519530 (441090,619490) | 146.36 (127.88,170.55) | 37374 (31832,43441) | 16.25 (13.81,18.93) | 1339828 (1132272,1583951) | 453.05 (387.49,524.65) |
| **GBD regions** | | | | | | | | |
| High-income Asia Pacific | 98890 (83696,116333) | 50.81 (43.28,59.61) | 239700 (207739,280017) | 121.95 (106.06,142.25) | 6067 (5506,6482) | 3.44 (3.07,3.69) | 132639 (122229,141110) | 69.35 (63.84,73.89) |
| High-income North America | 170213 (142093,200920) | 50.04 (41.83,59.11) | 416195 (351329,495267) | 121.5 (103.1,143.71) | 7370 (6673,7773) | 2.04 (1.85,2.15) | 156817 (146298,166869) | 45.32 (42.39,48.3) |
| Western Europe | 123808 (108996,141039) | 23.29 (20.2,26.49) | 259814 (227994,296840) | 49.52 (43.45,57.06) | 21275 (19677,22285) | 3.6 (3.33,3.77) | 388464 (368857,403548) | 68.98 (65.65,71.65) |
| Australasia | 7006 (6191,8018) | 30.4 (26.92,34.59) | 15735 (13746,17914) | 67.86 (59.26,77.24) | 1023 (928,1087) | 4.58 (4.13,4.88) | 18378 (17070,19421) | 79.92 (74.15,84.59) |
| Andean Latin America | 11033 (9384,12863) | 38.79 (34.09,44.19) | 24610 (20972,29128) | 84.05 (73.72,96.67) | 1557 (1339,1826) | 7.92 (6.79,9.29) | 43674 (37394,51016) | 182.66 (157.4,214.08) |
| Tropical Latin America | 40325 (33421,47560) | 38.07 (32,45.07) | 89074 (76220,104757) | 82.71 (70.73,96.97) | 3724 (3551,3851) | 4.38 (4.11,4.56) | 108767 (105210,112429) | 107.65 (103.36,111.49) |
| Central Latin America | 29759 (25987,33926) | 30.03 (26.61,33.8) | 60018 (52579,69308) | 56.03 (49.28,64.36) | 6331 (6047,6518) | 8.61 (8.13,8.89) | 159100 (154677,163221) | 176.68 (170.21,181.82) |
| Southern Latin America | 8277 (7177,9395) | 17.74 (15.47,20.07) | 17715 (15397,20185) | 37.6 (32.7,42.79) | 1264 (1178,1352) | 2.89 (2.68,3.09) | 29663 (27915,31466) | 64.49 (60.72,68.4) |
| Caribbean | 10310 (8903,11833) | 33.98 (29.82,38.63) | 23075 (19929,26955) | 74.29 (65.24,85.87) | 1659 (1453,1939) | 6.54 (5.76,7.63) | 47335 (40235,55119) | 169.4 (145.17,197.89) |
| Central Europe | 60933 (53386,70205) | 42.52 (37.29,48.78) | 142117 (124860,161642) | 98.31 (87.15,112.07) | 6920 (6633,7177) | 4.91 (4.7,5.1) | 179650 (173328,186268) | 123.77 (119.2,128.41) |
| Eastern Europe | 94747 (78000,113858) | 35.96 (29.76,42.48) | 218068 (185899,256023) | 82.15 (70.6,97.03) | 8990 (8692,9221) | 3.29 (3.17,3.37) | 275111 (265370,282509) | 101.14 (97.63,103.79) |

(Continued)

**Table 1.** (Continued)

| Character-istics | Incident case no. (95%UI) | ASIR/100,000 (95% UI) | Prevalence case no. (95%UI) | ASPR/100,000 (95% UI) | Deaths case no. (95%UI) | ASMR/100,000 (95% UI) | DALYs case no. (95%UI) | ASDR/100,000 (95% UI) |
|---|---|---|---|---|---|---|---|---|
| Central Asia | 19220 (16219,22201) | 33.94 (29.18,38.9) | 45148 (39248,52301) | 79.37 (70.21,90.79) | 1904 (1817,2007) | 3.91 (3.73,4.13) | 65049 (62002,68812) | 122.56 (116.77,129.47) |
| North Africa and Middle East | 89876 (72829,105835) | 38.39 (32.53,44.26) | 211248 (180245,254487) | 88.09 (77.21,103.39) | 9391 (7427,11637) | 5.99 (4.47,7.7) | 318032 (261074,375216) | 148.3 (119.71,180.67) |
| South Asia | 707562 (581838,831551) | 91.03 (76.61,106.58) | 1643607 (1408753,1949371) | 202.49 (176.09,236.74) | 96154 (78323,116317) | 16.66 (13.29,20.69) | 3313025 (2773093,3919543) | 464.96 (383.23,556.37) |
| Southeast Asia | 192161 (157226,225571) | 55.31 (46.81,64.61) | 474495 (409614,562015) | 134.25 (117.25,155.85) | 18038 (14511,23221) | 7.39 (5.88,9.39) | 586310 (480224,755010) | 192.86 (156.85,246.85) |
| East Asia | 741041 (612464,897969) | 73.45 (61.35,88.75) | 1782882 (1529682,2118579) | 176.33 (150.28,210.15) | 59396 (49631,70746) | 8.14 (6.82,9.76) | 1744242 (1466033,2047725) | 191.49 (161.05,225.7) |
| Oceania | 3873 (3193,4512) | 81.99 (69.59,94.5) | 9546 (8194,11199) | 197.2 (173.48,226.34) | 289 (232,374) | 11.04 (9.01,13.81) | 10169 (8096,13352) | 276.31 (223.83,354.67) |
| Western Sub-Saharan Africa | 75327 (61457,91335) | 50.78 (42.8,58.52) | 167825 (137779,209240) | 109.82 (94.4,129.77) | 7640 (6237,9222) | 8.73 (7.06,10.78) | 268965 (221828,322373) | 230.43 (188.42,276.78) |
| Eastern Sub-Saharan Africa | 55674 (44279,67696) | 41.36 (34.37,47.82) | 128023 (105979,157689) | 93.08 (80.67,109.53) | 11496 (6802,14893) | 13.49 (8.11,17.24) | 438933 (259324,577214) | 407.78 (238.8,526.4) |
| Central Sub-Saharan Africa | 17023 (13827,20337) | 43 (36.29,49.5) | 39106 (32584,47911) | 94.99 (82.47,110.37) | 2004 (1427,2645) | 10.21 (6.68,13.82) | 69754 (50829,88096) | 245.1 (171.9,325.55) |
| Southern Sub-Saharan Africa | 13358 (10819,15785) | 34.67 (29.25,40) | 30112 (25394,36770) | 75.86 (66.48,89.18) | 1380 (1172,1702) | 5.53 (4.62,6.99) | 40702 (34986,48655) | 130.58 (112.46,158.55) |

**Table 2. Numbers and age-standardized incidence, prevalence, deaths and DALYs rates of peptic ulcer disease in 2021.**

| Characteristics | Incident case no. (95%UI) | ASIR/100,000 (95% UI) | Prevalence case no. (95%UI) | ASPR/100,000 (95% UI) | Deaths case no. (95%UI) | ASMR/100,000 (95% UI) | DALYs case no. (95%UI) | ASDR/100,000 (95% UI) |
|---|---|---|---|---|---|---|---|---|
| Global | 2854370 (2438231,3264252) | 34.1 (29.12,38.97) | 6567782 (5798379,7597596) | 78.27 (69.02,90.75) | 230217 (193005,270858) | 2.75 (2.31,3.24) | 6057594 (5162099,7041854) | 71.56 (61.07,83.05) |
| SDI | | | | | | | | |
| High SDI | 510587 (441371,583786) | 30.97 (26.42,35.51) | 1222986 (1070914,1409189) | 74.82 (65.59,87.33) | 23775 (20824,25464) | 1.03 (0.92,1.09) | 473707 (435076,506842) | 24.55 (22.71,26.46) |
| High-middle SDI | 481239 (410252,560600) | 27.65 (23.58,31.87) | 1100182 (952267,1275701) | 63.1 (55.14,73.52) | 37481 (33689,41533) | 1.95 (1.75,2.15) | 847669 (774216,941432) | 45.08 (41.24,49.95) |
| Middle SDI | 850073 (722535,987507) | 31.87 (27.17,36.73) | 1952224 (1703650,2254015) | 72.45 (63.63,83.74) | 65033 (55949,77904) | 2.7 (2.31,3.22) | 1579710 (1389450,1868326) | 60.21 (52.85,71.06) |
| Low-middle SDI | 673338 (564738,770950) | 39.75 (34.05,45.45) | 1527859 (1330541,1784538) | 88.23 (77.9,102.01) | 69322 (52143,86311) | 5.17 (3.85,6.48) | 1982374 (1550828,2424753) | 127.95 (98.54,157.6) |
| Low SDI | 336786 (278693,398389) | 40.2 (34.52,45.53) | 759178 (646363,916362) | 87.98 (77.47,101.77) | 34365 (24980,41942) | 6.97 (4.97,8.55) | 1168176 (878955,1411988) | 178.38 (131.41,216.78) |
| GBD regions | | | | | | | | |
| High-income Asia Pacific | 110223 (95461,125870) | 37.38 (31.07,43.73) | 262794 (226577,303973) | 90.62 (76.91,107.94) | 4161 (3339,4635) | 0.68 (0.57,0.75) | 72898 (63552,80443) | 16.98 (15.2,19.14) |
| High-income North America | 215481 (183499,249825) | 38.42 (33.13,43.91) | 524119 (453897,605839) | 92.88 (80.75,107.74) | 4575 (4037,4878) | 0.68 (0.61,0.72) | 111311 (101789,122144) | 18.67 (17.11,20.55) |
| Western Europe | 98785 (86906,110648) | 13.8 (11.72,15.84) | 208443 (182883,238316) | 30.38 (26,35.57) | 9530 (8201,10287) | 0.86 (0.76,0.92) | 157891 (142167,168442) | 17.27 (15.9,18.35) |
| Australasia | 6006 (5098,6876) | 13.67 (11.35,15.9) | 12880 (10978,15103) | 30.01 (25.22,36.08) | 372 (317,408) | 0.61 (0.52,0.67) | 6233 (5511,6825) | 11.64 (10.38,12.75) |
| Andean Latin America | 12754 (11156,14474) | 20.42 (18.06,23.01) | 26973 (23501,31378) | 42.61 (37.27,49.31) | 1143 (942,1385) | 2 (1.65,2.42) | 25719 (21120,31073) | 42.78 (35.16,51.67) |
| Tropical Latin America | 42800 (36268,49486) | 16.88 (14.34,19.47) | 92746 (80858,107779) | 36.44 (31.86,42.44) | 5076 (4655,5423) | 2.02 (1.84,2.16) | 127425 (119490,136057) | 49.45 (46.29,52.83) |
| Central Latin America | 38762 (34069,43533) | 15.43 (13.65,17.28) | 75172 (65760,86198) | 29.56 (25.94,33.86) | 6425 (5681,7156) | 2.69 (2.38,3) | 139216 (123852,155632) | 55.76 (49.57,62.31) |
| Southern Latin America | 10527 (9199,11954) | 13 (11.25,14.85) | 22307 (19422,25687) | 27.79 (24.08,32.26) | 916 (826,990) | 1.02 (0.92,1.11) | 19216 (17730,20684) | 22.55 (20.82,24.33) |
| Caribbean | 12563 (10980,14291) | 24.96 (21.59,28.56) | 27199 (23583,31548) | 54.11 (46.6,62.94) | 1616 (1338,1961) | 3.01 (2.49,3.66) | 42010 (33910,52148) | 80.02 (64.59,99) |
| Central Europe | 63518 (56180,71761) | 35.61 (31.24,40.13) | 145032 (128065,163349) | 81.11 (71.86,92.58) | 7760 (7117,8265) | 3.49 (3.22,3.72) | 164866 (154031,176013) | 82.62 (77.23,88.34) |

*(Continued)*

**Table 2.** (Continued)

| Character-istics | Incident case no. (95%UI) | ASIR/100,000 (95% UI) | Prevalence case no. (95%UI) | ASPR/100,000 (95% UI) | Deaths case no. (95%UI) | ASMR/100,000 (95% UI) | DALYs case no. (95%UI) | ASDR/100,000 (95% UI) |
|---|---|---|---|---|---|---|---|---|
| Eastern Europe | 104073 (88322,122284) | 35.84 (30.51,41.38) | 238545 (204578,277278) | 81.64 (71.36,94.54) | 14517 (13360,15709) | 4.24 (3.91,4.59) | 359525 (332258,390690) | 112.63 (104.12,122.45) |
| Central Asia | 28551 (25078,32498) | 31.03 (27.46,35.02) | 65068 (57297,74410) | 70.3 (62.27,80.32) | 2596 (2311,2896) | 3.3 (2.93,3.68) | 79842 (71188,89669) | 89.89 (80.2,100.29) |
| North Africa and Middle East | 154212 (128371,181205) | 27.68 (23.69,31.84) | 357292 (300699,429589) | 62.67 (53.73,74.29) | 7728 (6513,9062) | 2.05 (1.74,2.4) | 214130 (180767,252189) | 44.98 (38.11,52.52) |
| South Asia | 677250 (566507,776408) | 39.78 (33.82,45.35) | 1509539 (1312046,1778513) | 86.41 (75.89,100.88) | 71469 (50012,94201) | 5.26 (3.65,6.99) | 2003927 (1457071,2580289) | 127.92 (92.21,165.49) |
| Southeast Asia | 252539 (212111,293575) | 35.96 (30.63,41.58) | 591043 (511914,694352) | 82.84 (72.07,96.87) | 20374 (17044,25524) | 3.53 (2.95,4.4) | 551384 (458398,693318) | 83 (69.62,103.69) |
| East Asia | 698169 (585315,833271) | 34.62 (29.35,40.37) | 1666223 (1435274,1923198) | 81.95 (71.21,95.06) | 41758 (32898,53012) | 2.16 (1.7,2.72) | 918377 (738327,1163522) | 45.1 (36.33,56.86) |
| Oceania | 7261 (6149,8393) | 65.4 (57.19,74.46) | 17278 (14914,20199) | 152.22 (133.42,174.34) | 397 (296,524) | 6.02 (4.63,7.73) | 13245 (9792,17428) | 149.91 (114.03,194.35) |
| Western Sub-Saharan Africa | 167873 (137236,203503) | 44.03 (37.89,50.76) | 375832 (308736,462906) | 95.44 (82.48,111.63) | 10899 (8823,12920) | 5.71 (4.78,6.71) | 384448 (303269,462094) | 141.59 (114.51,166.74) |
| Eastern Sub-Saharan Africa | 95488 (76419,114763) | 29.82 (25.31,34.22) | 219835 (183709,267755) | 66.85 (58.34,77.95) | 13101 (7992,17556) | 6.98 (4.15,9.34) | 489987 (308386,649943) | 198.29 (121.89,265.39) |
| Central Sub-Saharan Africa | 36651 (29945,44113) | 36.67 (31.76,41.81) | 83053 (69527,102197) | 79.59 (69.39,93.18) | 3308 (2165,4633) | 7.17 (4.24,10.15) | 106988 (75024,145305) | 162.47 (106.89,227.11) |
| Southern Sub-Saharan Africa | 20882 (17593,23900) | 30.53 (26.32,34.8) | 46408 (40490,54426) | 65.41 (57.82,75.73) | 2497 (2215,2855) | 4.96 (4.4,5.64) | 68956 (60119,79289) | 112.26 (99.26,128.42) |

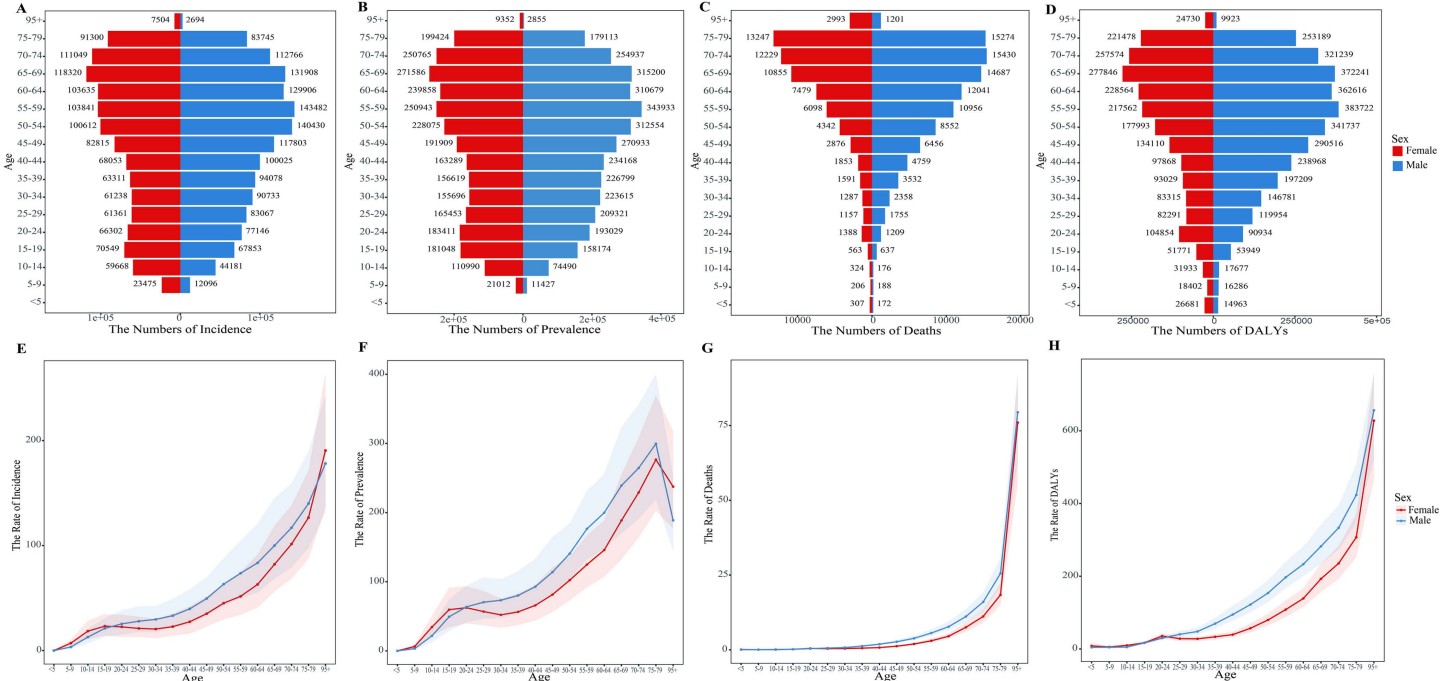

**Fig 1. Age-specific numbers and rates of PUD globally in 2021 for both sexes.** (A) Age-specific incidence number. (B) Age-specific prevalence number. (C) Age-specific mortality number. (D) Age-specific DALYs number. (E) Age-specific incidence rate. (F) Age-specific prevalence rate. (G) Age-specific mortality rate. (H) Age-specific DALYs rate.

APC = −1.34) (Figs 3B and 3F). The ASMR significantly decreased from 1990 to 2021 in both males (1990–1994 APC=−1.82, 1994–2001 APC= −3.50, 2001–2004 APC = −2.86, 2004–2007 APC=−4.95, 2007–2012 APC=−3.76, 2012–2021 APC=−3.23) and females (1990–1995 APC=−1.55, 1995–2000 APC= −3.05, 2000–2004 APC=−1.81, 2004–2011 APC= −3.72, 2011–2021 APC = −2.52) (Figs 3C and 3G). The ASDR exhibited a significant decline from 1990 to 2021 for both males (1990–1994 APC=−1.77, 1994–2004 APC= −3.54, 2004–2007 APC = −4.72, 2007–2013 APC=−4.05, 2013–2021 APC=−3.24) and females (1990–1994 APC=−1.44, 1994–2004 APC= −2.59, 2004–2011 APC=−3.82, 2011–2021 APC= −2.70) (Figs 3D and 3H). The AAPCs of ASIR, ASPR, ASMR and ASDR for PUD decreased by −1.65 (95% CI: −1.69, −1.61), −1.69 (95% CI: −1.74, −1.63), −3.02 (95% CI: −3.13, −2.91) and −3.17 (95% CI: −3.24,-3.10) from 1990 to 2021 on a global scale, respectively. Surprisingly, males exhibited a lower AAPC in incidence, prevalence, mortality and DALYs rates compared to females (S1 Table in S1 File).

Fig 4 illustrated the observed national and regional ASIR, ASPR, ASMR and ASDR in relation to the SDI, compared to the expected levels for each location based on SDI. From 1990 to 2021, ASRs of incidence and prevalence of PUD initially increased but began to decline once the SDI exceeded 0.4, and then started to rise steadily at SDI values around 0.65. The South Asia, North Africa and Middle East closely followed the anticipated trends throughout the study period (Figs 4A and 4B). In contrast, the negative associations were found between the ASMR, ASDR and SDI from 1990 to 2021. The Global, Western Europe, Caribbean, Andean Latin America, Central Latin America, Southeast Asia and Oceania closely adhered to the anticipated trends over the study period (Figs 4C and 4D). In 2021, a negative correlation was observed between the ASIR, ASPR, and SDI for PUD at the national level, although a few exceptions were noted (Figs 5A and 5B). Similar patterns were also observed for ASMR and ASDR in relation to SDI (Figs 5C and 5D).

**Fig 2. Trends in the all-age numbers and age-standardized incidence, prevalence, mortality and DALYs rates of PUD by sex from 1990 to 2021.** (A) Incidence number and rate. (B) Prevalence number and rate. (C) Mortality number and rate. (D) DALYs number and rate.

Across all countries and and territories, in 2021, the Kiribati had the global highest ASIR (118.12 (95% UI:102.39–138.58) per 100,000), while Israel had the lowest ASIR (5.42 (95% UI: 4.47–6.43) per 100,000). The Taiwan (335.31 (95% UI: 284.75–399.55) per 100,000) had the highest ASPR, while Israel (10.73 (95% UI: 8.66–12.97) per 100,000) had the lowest ASPR in 2021. The global highest ASMR (25.41 (95% UI: 14.98–38.59) per 100,000) and ASDR (552.24 (95% UI: 327.15–868.73) per 100,000) of PUD were seen in Cambodia, whereas the lowest ASMR (0.19 (95% UI: 0.14–0.26) per 100,000) and ASDR (4.3 (95% UI: 3.23–5.48) per 100,000) were observed in Andorra (Tables 3 and 4).

We employed the ARIMA model to analyze the ASIR, ASPR, ASMR and ASDR of PUD for both sexes from 1990 to 2021, with predictions extending to 2040 (Fig 6). The PUD ASIR was projected to decrease from 37.45 per 100,000 in 2021 to 32.02 per 100,000 in 2040 for males, and from 30.88 per 100,000 in 2021 to 20.15 per 100,000 in 2040 for females (Figs 6A and 6E, and S2-3 Tables in S1 File). The PUD ASPR was expected to decline from 85.69 per 100,000 in 2021 to 67.05 per 100,000 in 2040 for males, and from 71.14 per 100,000 in 2021 to 45.43 per 100,000 in 2040 for

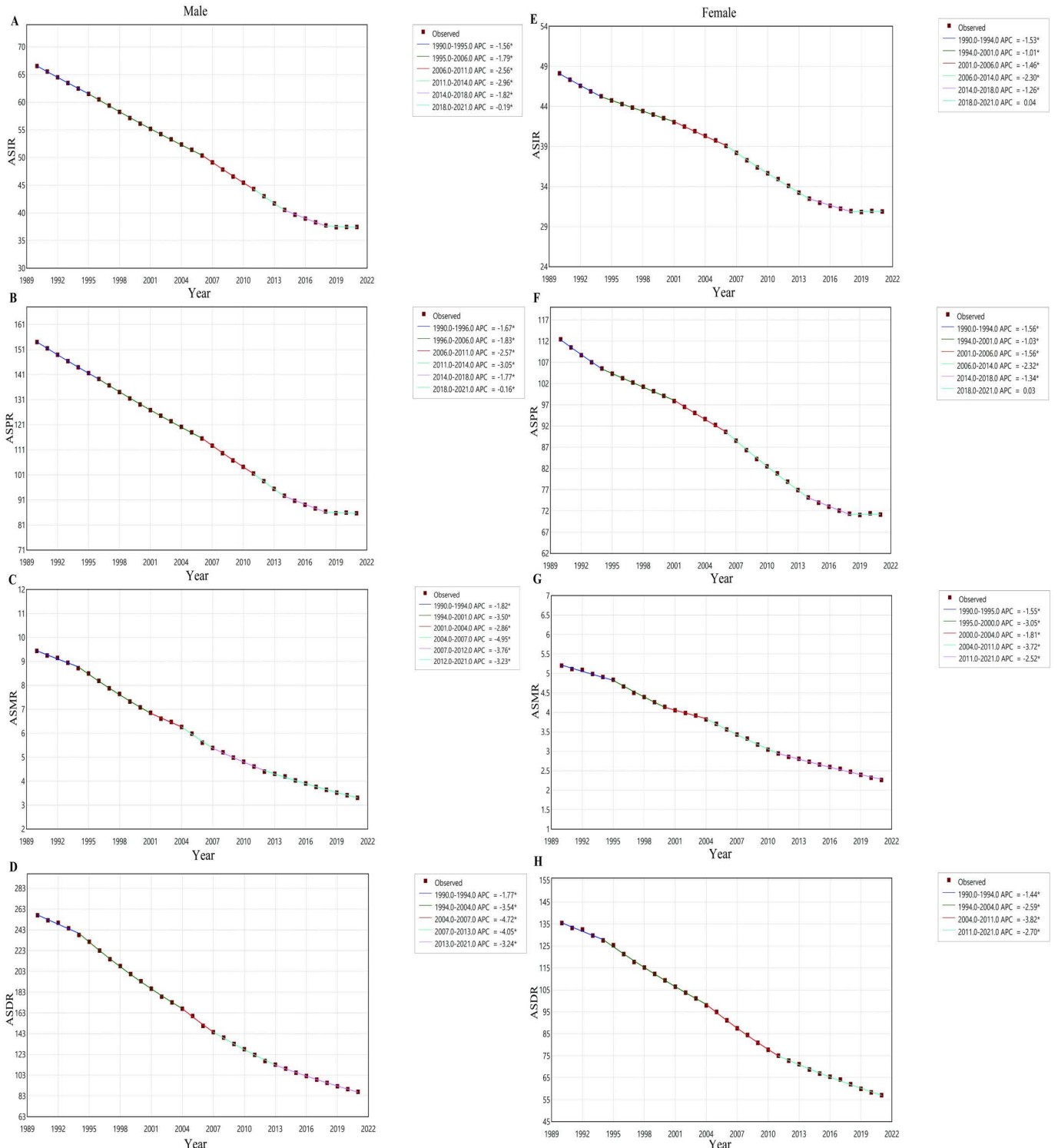

* Indicates that the Annual Percent Change (APC) is significantly different from zero at the alpha=0.05 level.
* Indicates that the Annual Percent Change (APC) is significantly different from zero at the alpha=0.05 level.

**Fig 3. Joinpoint regression analysis of the sex-specific ASRs for PUD globally from 1990 to 2021.** (A) ASIR for males. (B) ASPR for males. (C) ASMR for males. (D) ASDR for males. (E) ASIR for females. (F) ASPR for females. (G) ASMR for females. (H) ASDR for females.

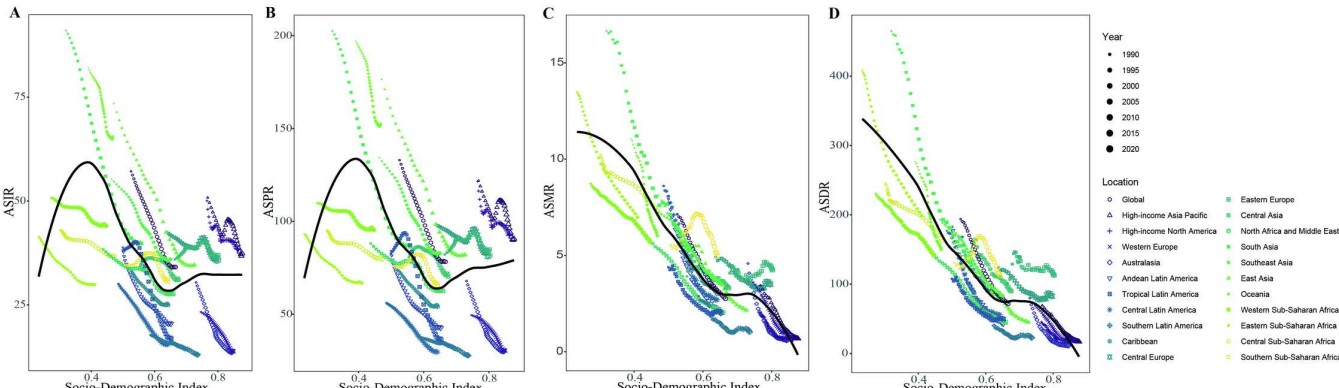

**Fig 4. Age-standardized rates of PUD globally and for 21 GBD regions by SDI, 1990–2021.** (A) Age-standardized incidence rates. (B) Age-standardized prevalence rates. (C) Age-standardized mortality rates. (D) Age-standardized DALYs rates. (For each region, the points arranged from left to right represent estimates for each year from 1990 to 2021).

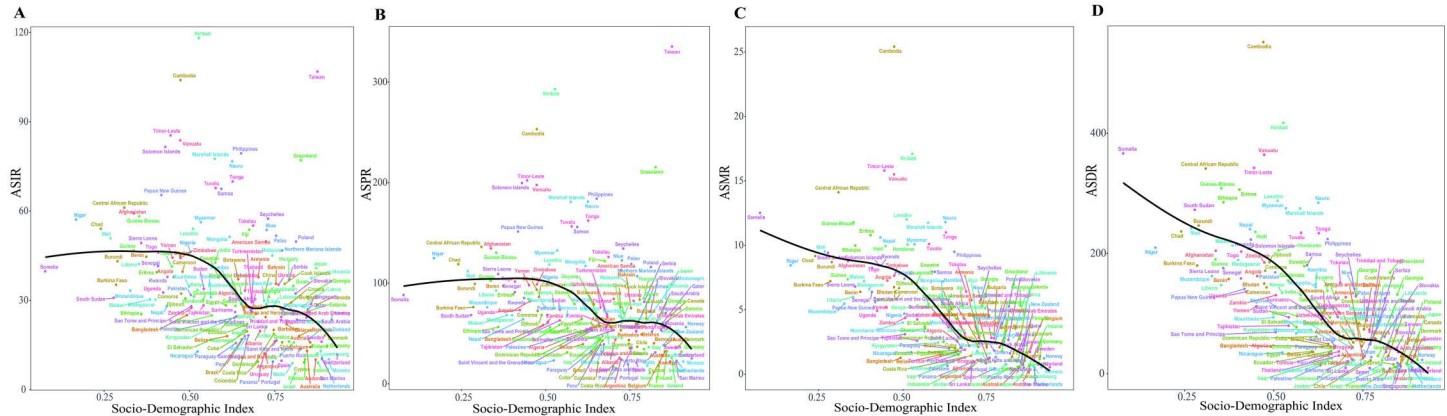

**Fig 5. Age-standardized rates of PUD in 204 countries globally by SDI in 2021.** (A) Age-standardized incidence rates. (B) Age-standardized prevalence rates. (C) Age-standardized mortality rates. (D) Age-standardized DALYs rates. (The blank lines represent the expected ASRs based on the SDI).

**Table 3. The top four countries or territories with the highest ASRs of PUD in 2021.**

| Characteristics | ASIR/100,000 (95% UI) | Characteristics | ASPR/100,000 (95% UI) | Characteristics | ASMR/100,000 (95% UI) | Characteristics | ASDR/100,000 (95% UI) |
|---|---|---|---|---|---|---|---|
| Kiribati | 118.12 (102.39,138.58) | Taiwan | 335.31 (284.75,399.55) | Cambodia | 25.41 (14.98,38.59) | Cambodia | 552.24 (327.15,868.73) |
| Taiwan | 106.82 (86.41,128.42) | Kiribati | 292.98 (258.62,332.56) | Kiribati | 17.1 (12.47,24.09) | Kiribati | 417.24 (305.55,583.12) |
| Cambodia | 104.01 (90.78,121.31) | Cambodia | 253.17 (221.37,287.89) | Timor-Leste | 15.8 (10.3,23.74) | Somalia | 366.53 (171.1,678.45) |
| Laos | 94.53 (82.05,113.15) | Laos | 229.86 (200.98,262.6) | Vanuatu | 15.5 (10.77,22.29) | Vanuatu | 364.34 (255.74,504.07) |

**Table 4. The top four countries or territories with the lowest ASRs of PUD in 2021.**

| Characteristics | ASIR/100,000 (95% UI) | Characteristics | ASPR/100,000 (95% UI) | Characteristics | ASMR/100,000 (95% UI) | Characteristics | ASDR/100,000 (95% UI) |
|---|---|---|---|---|---|---|---|
| Israel | 5.42 (4.47,6.43) | Israel | 10.73(8.66,12.97) | Andorra | 0.19 (0.14,0.26) | Andorra | 4.3 (3.23,5.48) |
| Panama | 7.88 (6.82,8.9) | Panama | 15.26 (12.89,17.96) | Indonesia | 0.29 (0.24,0.34) | San Marino | 6.23(4.54,8.27) |
| Malta | 8.89 (7.53,10.23) | Costa Rica | 17.33(15.08,20.07) | Puerto Rico | 0.32(0.27,0.38) | Israel | 7.53 (6.7,8.3) |
| Iceland | 9.03(7.33,10.62) | Cyprus | 17.71 (14.74,20.99) | San Marino | 0.34 (0.23,0.46) | Indonesia | 8.35 (6.91,9.82) |

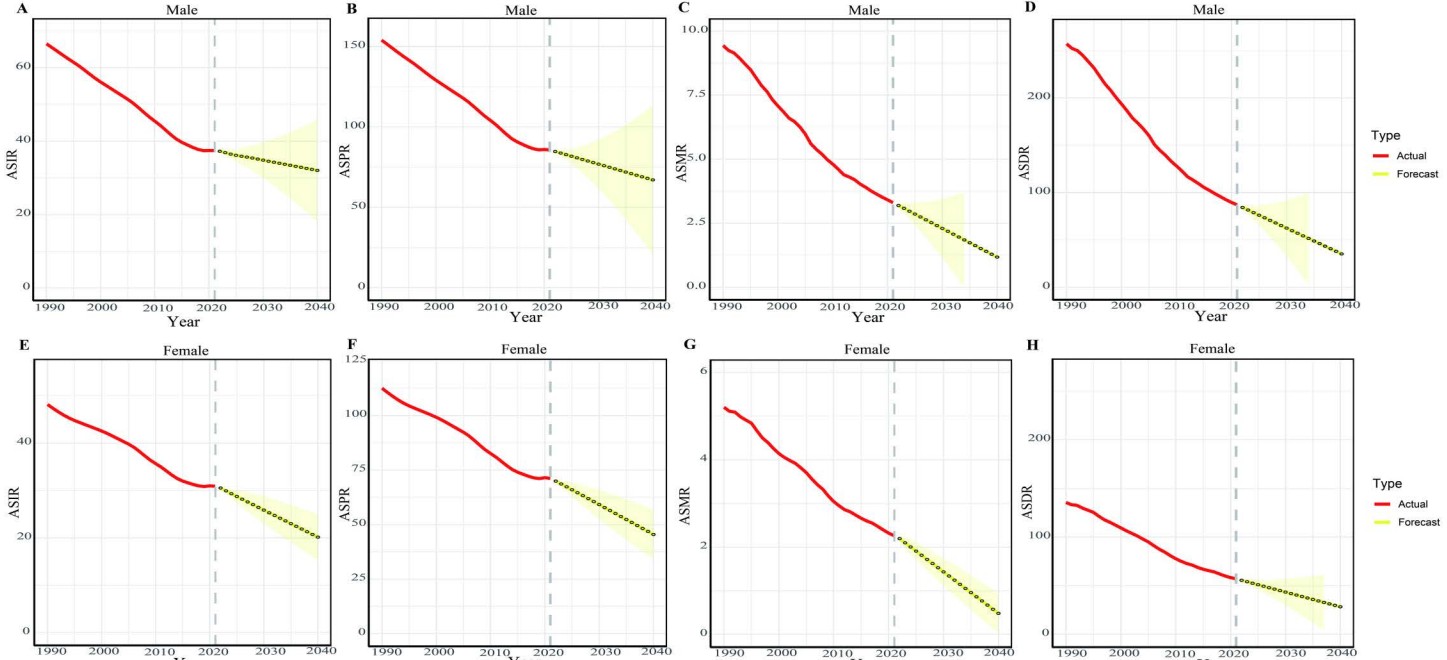

**Fig 6. Predicted trends in Age-standardized rates of PUD for both sexes globally over the next 19 years (2022-2040).** (A) ASIR for males. (B) ASPR for males. (C) ASMR for males. (D) ASDR for males. (E) ASIR for females. (F) ASPR for females. (G) ASMR for females. (H) ASDR for females. (The red lines illustrate the actual trend of PUD ASRs from 1990 to 2021, while the yellow dotted lines and shaded regions indicate the predicted trend along with its 95% CI).

females (Figs 6B and 6F, and S4-5 Tables in S1 File). The PUD ASMR was projected to decrease from 3.31 per 100,000 in 2021 to 1.18 per 100,000 in 2040 for males. For females, it was expected to decline from 2.26 per 100,000 in 2021 to 0.48 per 100,000 in 2040 (Figs 6C and 6G, and S6-7 Tables in S1 File). The PUD ASDR was expected to decline from 87.02 per 100,000 in 2021 to 35.24 per 100,000 in 2040 for males, and from 57.05 per 100,000 in 2021 to 28.26 per 100,000 in 2040 for females (Figs 6D and 6H, and S8-9 Tables in S1 File).

### Risk factor estimation for PUD

We found that smoking was the predominant risk factor for PUD in both sexes based on the GBD risk factor data. Subsequently, the percentage contribution of smoking to PUD DALYs and deaths by gender, SDI quintile, and GBD region in 1990 and 2021 were shown in Fig 7. Across all regions,

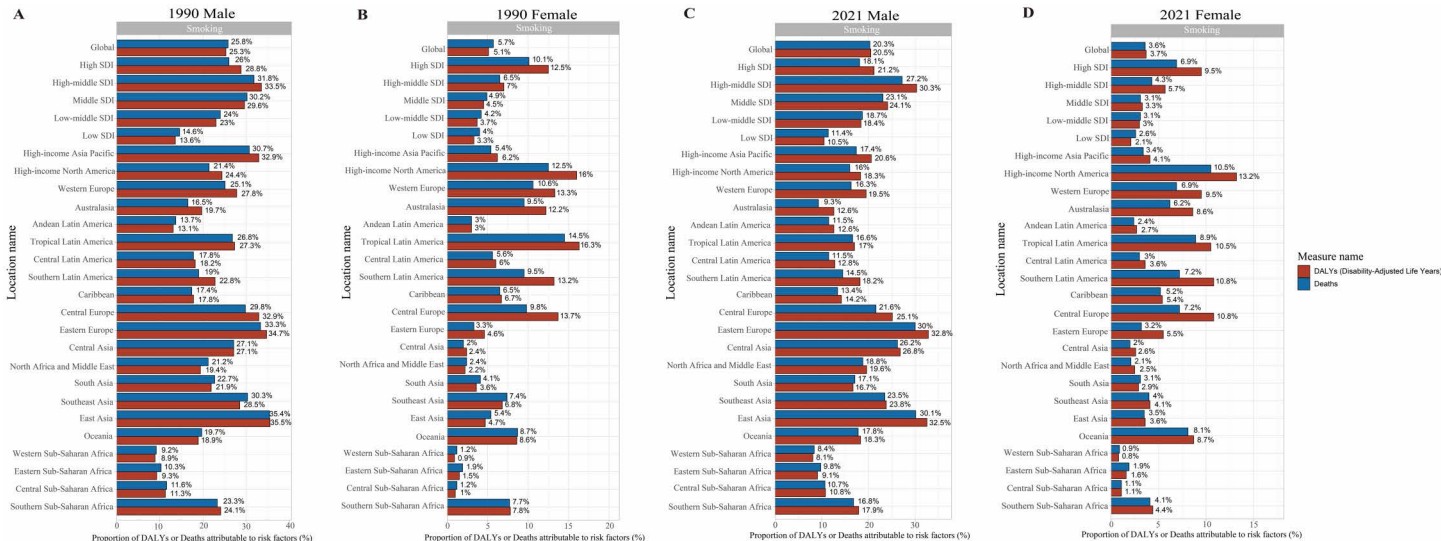

**Fig 7. Proportion of PUD DALYs and deaths attributable to smoking by gender, SDI quintile, and GBD region in 1990 and 2021.** (A) Proportion of PUD DALYs and deaths attributable to smoking for male by SDI quintile and GBD region in 1990. (B) Proportion of PUD DALYs and deaths attributable to smoking for female by SDI quintile and GBD region in 1990. (C) Proportion of PUD DALYs and deaths attributable to smoking for male by SDI quintile and GBD region in 2021. (D) Proportion of PUD DALYs and deaths attributable to smoking for female by SDI quintile and GBD region in 2021.

smoking contributed a higher percentage to PUD DALYs and deaths in males compared to females (Fig 7). From 1990 to 2021, the percentage contribution of smoking to PUD DALYs and deaths decreased for both sexes. However, the impact of smoking varied by region. For example, the impact of smoking was highest in East Asia (32.5% of DALYs and 30.1% of deaths were attributable to smoking) and Eastern Europe (32.8% and 30.0%), where smoking is still prevalent, and lowest in western sub-Saharan Africa (8.1% and 8.4%) for males in 2021. Similarly, the impact of smoking was highest in High-income North America (13.2% of DALYs and 10.5% of deaths were attributable to smoking), and lowest in western sub-Saharan Africa (0.8% and 0.9%) for females in 2021.

## Discussion

PUD is typically defined as a rupture of the gastric or duodenal mucosa greater than 3–5 mm, which arises from an imbalance between protective and damaging factors of the gastric and duodenal mucosa [30]. In this study, we presented the most current and comprehensive assessment of the global, regional, and national disease burden of PUD from 1990 to 2021. Although the ASIR and ASPR of PUD declined globally in 2021, the absolute number increased, a finding consistent with previous studies [31,32]. We speculated that the above phenomenon may be attributed to several factors, including population growth, advancements in diagnostic techniques, the use of proton pump inhibitors (PPIs), and the widespread administration of anti-H. pylori treatments [4,33]. However, the numbers and ASRs of deaths and DALYs of PUD decreased between 1990 and 2021 due to better prevention and therapy [34,35]. As for age and gender differences, men exhibited higher numbers and ASRs of incidence, prevalence, deaths, and DALYs associated with PUD than women across most age cohorts in 2021, which indicated that the PUD burden was significantly heavier in males. Estrogen may play a protective role against ulcers by inhibiting both the synthesis and release of gastrin, as well as reducing gastric acid secretion [36,37]. Meanwhile, poor lifestyle habits such as smoking and alcohol consumption are more prevalent among men and are recognized as significant risk factors for PUD [14,38,39]. The aforementioned references may partially account for the higher numbers and ASRs of PUD in men. From 1990 to 2021, the ASRs of incidence, prevalence, deaths

and DALYs for PUD have significantly declined for both sexes. The decreasing trends observed in males were more pronounced than those in females. However, the rates among males still remained higher. The results of the joinpoint regression analyses were consistent with the above trends. Previous research has shown that the prevalence of active smoking among men decreased from 60.0% to 47.6%, while the proportion of daily alcohol consumption declined from 19.0% to 13.1% between 1997 and 2020, which could partly explain the aforementioned decreasing trends for males [40]. Futhermore, the decreasing trends observed for both sexes have plateaued in recent years, possibly due to a shift in the primary cause of ulcers in numerous countries from H. pylori infection to the use of NSAIDs [32,41,42].

When stratified by SDI, the ASIR and ASPR of PUD initially increased but began to decline once the SDI exceeded 0.4, and then started to rise steadily at SDI values around 0.65 from 1990 to 2021. In lower SDI regions, the ASIR and ASPR of PUD showed upward trends with increasing SDI, which is closely associated with the high H. pylori infection rates and suboptimal medical treatment [43,44]. Meanwhile, the results also demonstrated a gradual increase in the ASIR and ASPR of PUD over time in high SDI regions. Bariatric surgery has been a proper therapy to reduce weight and comorbidities in high SDI regions [45,46]. However, marginal ulcers occurring at or distal to the gastroenteral anastomosis were observed in approximately 5% of obese patients who undergo gastric bypass surgery [47,48]. Moreover, the incidence of these ulcers may rise to 27%−36% in patients with upper gastrointestinal symptoms following gastric bypass surgery, which could explain the upward trends in ASIR and ASPR of PUD in high SDI regions [49,50]. Additionally, the ASMR and ASDR associated with PUD decreased across all groups from 1990 to 2021, likely reflecting increased awareness and improved treatment of H. pylori infection, along with better management of other chronic conditions. In 2021, the ASIR, ASPR, ASMR and ASDR of PUD generally decreased with increasing SDI across various countries and regions. However, in certain areas, such as Kiribati, Taiwan, Cambodia, Timor-Leste, and Vanuatu, the ASRs of PUD still remained abnormally high. This phenomenon may be associated with elevated rates of H. pylori infection, Chronic Urticaria and the ethnic characteristics of islanders [32,51]. We also found that Israel had the lowest ASIR and ASPR, while Andorra reported the lowest ASMR and ASDR of PUD in 2021, which appeared to be associated with limited access to diagnostic services [52–54]. Tracking disease outbreaks and forecasting trends is a crucial aspect of disease prevention and management. Based on the ARIMA model, we projected that the global ASIR, ASPR, ASMR, and ASDR for PUD will decline by 2040 when compared to the figures from 2021. However, over the next nearly two decades, the decline in ASIR and ASPR for females became steeper compared to males. We hypothesized that the observed trends in gender differences may be linked to certain gender-related risk factors, warranting further investigation. In our risk factor analysis, we identified smoking as the primary risk factor associated with the DALYs and deaths for PUD for both sexes in 1990 and 2021. Meanwhile, PAF for smoking was consistently higher in men than in women across all regions. Previous studies have established that smoking was a significant risk factor for PUD, highlighting its influence on both the onset and progression of this condition [55–57].

There were some limitations in this study. Firstly, given that the data from the GBD database were collected from various locations globally, there may be constraints related to underdiagnosis and underreporting, potentially leading to an underestimate of our results. Secondly, there are various types of PUD, but the GBD database does not account for this complexity, focusing solely on whether the population falls under the PUD classification. Thirdly,

while the ARIMA model effectively handles time series complexities, it may not account for all influencing factors, such as socioeconomic shifts, medical advancements or changes in patient behavior, etc. The projections should be viewed as potential scenarios rather than definitive outcomes. To ensure accuracy and reliability, the ARIMA model should be regularly updated to incorporate new data and trends, and periodic adjustments and recalibrations are required. Fourthly, the GBD encompasses a limited number of risk factors, which inevitably leads to the oversight of certain significant factors that may impact the burden of PUD. Thus, we will provide a more comprehensive analysis of risk factors for PUD as new data becomes available in the future. However, it is important to note that the PAF reflects an estimated burden assuming a causal relationship, which in the context of the GBD study is based on prior evaluation of causal criteria. Therefore, PAF

use in this study does not itself establish causality, but quantifies the impact of a risk factor where causal inference has already been determined by the GBD methodology.

## Conclusion

In summary, our research examined the epidemiological characteristics of PUD across various countries and territories, as well as among different age groups and both sexes. Despite a decline in the global burden of PUD in recent decades, significant regional disparities still persist. The burden remains particularly high in low-SDI countries, underscoring the urgent need for targeted interventions. Meanwhile, the variation in PUD burden across different sexes and age groups highlights the need for tailored prevention and treatment strategies to address the specific needs of these populations. Moreover, smoking was identified as the primary risk factor for PUD based on GBD 2021 data, and PAF values for smoking showed area- and sex-specific differences. Therefore, intensive reduction of tobacco use is strongly recommended. Where possible, collecting country-level data on additional risk factors would be valuable. Briefly, enhancing global and regional collaboration, improving access to diagnostic, treatment, and prevention services, and creating targeted interventions for high-risk groups are key steps in reducing the impact of PUD.

## Supporting information

**S1 File.** S1 Table. Global trends for ASRs of PUD wordwide from 1990 to 2021 by Joinpoint regression analysis. S2 Table. Predicted trends of PUD ASIR for males globally over the next 19 years. S3 Table. Predicted trends of PUD ASIR for females globally over the next 19 years. S4 Table. Predicted trends of PUD ASPR for males globally over the next 19 years. S5 Table.Predicted trends of PUD ASPR for females globally over the next 19 years. S6 Table. Predicted trends of PUD ASMR for males globally over the next 19 years. S7 Table. Predicted trends of PUD ASMR for females globally over the next 19 years. S8 Table. Predicted trends of PUD ASDR for males globally over the next 19 years. S9 Table. Predicted trends of PUD ASDR for females globally over the next 19 years.
(ZIP)

## Acknowledgments

We gratefully acknowledge all staff for their contributions to the GBD database.

## Author contributions

**Conceptualization:** Wende Hao.

**Data curation:** Wende Hao.

**Formal analysis:** Wende Hao.

**Methodology:** Wende Hao, Chaoyue Zheng.

**Software:** Wende Hao, Chaoyue Zheng.

**Supervision:** Zhenjun Wang, Huachong Ma.

**Writing – original draft:** Wende Hao.

**Writing – review & editing:** Huachong Ma.

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
