## [Decision Letter · Decision Letter 0]

PONE-D-24-46593Global burden and risk factors of peptic ulcer disease between 1990 and 2021: an analysis from the Global Burden of Disease Study 2021PLOS ONE

Dear Dr. Ma,

Thank you for submitting your manuscript to PLOS ONE. After careful consideration, we feel that it has merit but does not fully meet PLOS ONE’s publication criteria as it currently stands. Therefore, we invite you to submit a revised version of the manuscript that addresses the points raised during the review process.

We look forward to receiving your revised manuscript.

Kind regards,

Emmanuel O Adewuyi, BPharm, MPH, PhD

Academic Editor

PLOS ONE

Journal Requirements:

5. We note that Figure 5 in your submission contain map/satellite images which may be copyrighted. All PLOS content is published under the Creative Commons Attribution License (CC BY 4.0), which means that the manuscript, images, and Supporting Information files will be freely available online, and any third party is permitted to access, download, copy, distribute, and use these materials in any way, even commercially, with proper attribution. For these reasons, we cannot publish previously copyrighted maps or satellite images created using proprietary data, such as Google software (Google Maps, Street View, and Earth). For more information, see our copyright guidelines: http://journals.plos.org/plosone/s/licenses-and-copyright.

 a. You may seek permission from the original copyright holder of Figure 5 to publish the content specifically under the CC BY 4.0 license. 

Additional Editor Comments:

The reviewers have provided feedback on this manuscript, highlighting several key issues. I invite the authors to revise their manuscript and respond comprehensively to all reviewer comments.

One reviewer has noted a similar publication on the same subject. The authors should clearly distinguish their work from the existing study, emphasizing how their research advances or builds upon previous findings.

Additionally, the reviewers have recommended improving the presentation of the manuscript, including grammatical refinements. Strengthening the literature review in the introduction section and expanding the discussion, including appropriate insight on the differences observed between male and female** ** would further enhance the quality of the work**.**

Reviewers' comments:

Reviewer's Responses to Questions

**Comments to the Author**

1. Is the manuscript technically sound, and do the data support the conclusions?

Reviewer #1: Yes

Reviewer #2: Partly

Reviewer #3: Partly

2. Has the statistical analysis been performed appropriately and rigorously? 

Reviewer #1: Yes

Reviewer #2: I Don't Know

Reviewer #3: Yes

3. Have the authors made all data underlying the findings in their manuscript fully available?

Reviewer #1: Yes

Reviewer #2: Yes

Reviewer #3: Yes

4. Is the manuscript presented in an intelligible fashion and written in standard English?

Reviewer #1: Yes

Reviewer #2: Yes

Reviewer #3: No

5. Review Comments to the Author

Reviewer #1: Revisão PLOS ONE - PONE-D-24-46593 Research Article

Global burden and risk factors of peptic ulcer disease between 1990 and 2021: an analysis from the Global Burden of Disease Study 2021.

The manuscript is well written, the topic is highly relevant and the references used are current. However, I was disturbed by the large number of abbreviations in the text, and I had the impression that there was a lack of better discussion of the results found. Overall, this is promising work on a topic that is of great relevance to the population.

Major issues:

According to the submission guidelines:

Define abbreviations upon first appearance in the text. Do not use non-standard abbreviations unless they appear at least three times in the text. Keep abbreviations to a minimum.

Introduction

Few references in the introduction, the topic should be explored better in the literature.

Discussion

I had the impression that a better discussion of the results found was missing. The discussion is not a summary of the results. You should compare the results obtained with other results available in the literature and find associations that corroborate the data found.

Minor issues:

Correct minor errors according to the submission guidelines:

- Include page numbers and line numbers in the manuscript file. Use continuous line numbers (do not restart the numbering on each page).

- Figure captions are inserted immediately after the first paragraph in which the figure is cited. Figure files are uploaded separately.

- Use figure label with Arabic numerals, and “Figure” abbreviated to “Fig” (e.g. Fig 1, Fig 2, Fig 3, etc).

Materials and methods

Add the website as a reference instead of placing it in the middle of the text (https://vizhub.healthdata.org/gbd-results/).

Discussion

Please find a more recent reference to replace reference number 19.

"which indicated that the PUD burden was significantly heavier in males, potentially due to lower levels of estrogen" - This assumption needs a reference or better argumentation.

"which appeared to be associated with limited access to diagnostic services." - A reference is needed for this assumption.

Reviewer #2: Dear authors, congratulations on your significant contribution to advancing our understanding of the epidemiological characteristics and risk factors associated with peptic ulcer disease. Here are my comments:

Q0: A similar study using the same material was published in 2023: "Peptic ulcer disease burden, trends, and inequalities in 204 countries and territories, 1990–2019: a population-based study" ( doi: 10.1177/17562848231210375). Beyond sex (male patients being more vulnerable in terms of incidence, prevalence mortality and DALY) and smoking (a primary risk factor), were there any other notable findings comparing the 2023 study to the current one? Additionally, did the COVID-19 pandemic influence the epidemiological characteristics of peptic ulcer disease between 2020 and 2021? If not, what are the authors' perspectives on this

Q1: The authors noted in the Discussion section that 'the ASRs of PUD remain abnormally high in Taiwan.' However, in the 2023 study mentioned in Q0 stated that 'EAPCs of ASDR were significantly decreasing in ... Taiwan'. What are the authors' opinions on this?

Q2: The ASRs of incidence, prevalence, deaths and DALYs of PUD decrease more rapidly in males than in females. Please discuss the possible reasons for this phenomenon.

Q3: How can the causal relationship between peptic ulcer disease (PUD)-related disability-adjusted life years (DALYs) and cigarette smoking be determined?

Q4: In "Global and regional burden of PUD", the authors mentioned: "The Taiwan (Province of China) (335.31 (95% UI: 284.75 to 399.55) per 100,000) had the highest ASPR." Though this annotation '(Province of China)' was shown in IHME forms, Taiwan is not globally recognized as a province of China; if it were, China would not pursue its annexation. For instance, does China plan to annex Hebei or Guangdong (both Province of China) in the future? Please remove the annotation '(Province of China)' to avoid potential political issues in scientific research.

Reviewer #3: Dear Editor and Authors,

This manuscript presents an analysis of the global burden of peptic ulcer disease (PUD) from 1990 to 2021, using data from the Global Burden of Disease (GBD) Study 2021. The study examines incidence, prevalence, mortality, and disability-adjusted life years (DALYs), and explores the association with potential risk factors. The topic is relevant, given the clinical and economic impact of PUD worldwide. While the study has merit, there are several areas that require attention before it can be considered for publication.

The introduction, while providing necessary background, suffers from a lack of specific focus and a strong "hook" to immediately engage the reader. It lacks a clearly stated hypothesis and a concise explanation of the study's novel contributions to the existing literature. Furthermore, the objectives are stated too broadly. To improve, provide a brief "roadmap" of the paper's structure.

The manuscript uses an ARIMA model to predict PUD burden until 2050. Long-term epidemiological predictions are inherently unreliable due to numerous unpredictable factors (e.g., changes in healthcare policy, emergence of new treatments, shifts in risk factor prevalence, and unforeseen global events). The manuscript fails to adequately acknowledge and address these limitations.

The conclusion that smoking is the only primary risk factor associated with DALYs and deaths from PUD is an overstatement not supported by a comprehensive analysis. The authors present this conclusion without exploring potential confounding factors or interactions between different risk factors.

The manuscript reports gender differences in PUD burden but fails to provide a meaningful explanation for these disparities. Simply stating that males have higher rates than females is insufficient. The authors do not explore the underlying biological, behavioral, or socioeconomic factors that could explain these differences.

The conclusion states that "precise prevention strategies are essential to mitigate the disease burden of PUD," but this statement is generic and lacks specific, actionable recommendations. The manuscript does not provide a clear link between the study's findings and concrete policy implications. The authors do not discuss how the observed trends and risk factors should inform the development and implementation of targeted prevention programs.

Finally, while the language is generally understandable, the manuscript contains several minor grammatical and stylistic errors that detract from its overall clarity and professionalism. A thorough proofreading by a native English speaker is recommended to address these issues and ensure the manuscript meets the standards for publication.

6. PLOS authors have the option to publish the peer review history of their article (what does this mean? ). If published, this will include your full peer review and any attached files.

**Do you want your identity to be public for this peer review?** For information about this choice, including consent withdrawal, please see our Privacy Policy .

Reviewer #1: No

Reviewer #2: No

Reviewer #3: No

---

## [Author Response · Author response to Decision Letter 1]

23 Mar 2025

Dear editor & reviewers,

Thank you for reviewing our manuscript and for the constructive comments, which greatly helped us to improve the manuscript. We have revised the manuscript in accordance with your comments. And point-by-point responses to the comments were as follows:

Journal Requirements:

1.Please ensure that your manuscript meets PLOS ONE's style requirements, including those for file naming.

Answer: Thank you for your comment and providing good suggestion for our manuscript. We have revised the manuscript based on PLOS ONE's style requirements, including those for file naming.

Answer: Thank you for your comment and providing good suggestion for our manuscript. We have removed the funding-related text from the manuscript.

Answer: Thank you for your comment and providing good suggestion for our manuscript. We have corrected the fund information in the ‘Funding Information’ section.

4.PLOS requires an ORCID iD for the corresponding author in Editorial Manager on papers submitted after December 6th, 2016. Please ensure that you have an ORCID iD and that it is validated in Editorial Manager. To do this, go to ‘Update my Information’ (in the upper left-hand corner of the main menu), and click on the Fetch/Validate link next to the ORCID field. This will take you to the ORCID site and allow you to create a new iD or authenticate a pre-existing iD in Editorial Manager.

Answer: Thank you for your comment and providing good suggestion for our manuscript. We have authenticated a pre-existing iD in Editorial Manager.

5. We note that Figure 5 in your submission contain map/satellite images which may be copyrighted. All PLOS content is published under the Creative Commons Attribution License (CC BY 4.0), which means that the manuscript, images, and Supporting Information files will be freely available online, and any third party is permitted to access, download, copy, distribute, and use these materials in any way, even commercially, with proper attribution. For these reasons, we cannot publish previously copyrighted maps or satellite images created using proprietary data, such as Google software (Google Maps, Street View, and Earth). For more information, see our copyright guidelines: http://journals.plos.org/plosone/s/licenses-and-copyright.

Answer: Thank you for your comment and providing good suggestion for our manuscript. We have replaced Figure 6 with Tables 3 and 4.

Answer: Thank you for your comment and providing good suggestion for our manuscript. We have modified the content of the Supporting Information according to the guidelines.

Reviewer #1:

(Please refer to the line numbers of the revised version of the manuscript)

Major issues:

1.According to the submission guidelines: Define abbreviations upon first appearance in the text. Do not use non-standard abbreviations unless they appear at least three times in the text. Keep abbreviations to a minimum.

Answer: Thank you for your comment and providing good suggestion for our manuscript. We have deleted the unnecessary acronyms (Page7 line187, line195-196) and spelled out some of the acronyms (Page3 line66) according to the submission guidelines.

2.Introduction: Few references in the introduction, the topic should be explored better in the literature.

Answer: Thank you for your comment and providing good suggestion for our manuscript. We have added related references in the introduction section (Page3 line85; Page4 line90, line 93, line 96, line 100, line 103, line 106 and line 110). Meanwhile, relevant content has been added to the introduction to improve the presentation of the manuscript(Page4 line104-106 and line110; Page5 line125-126).

3.Discussion: I had the impression that a better discussion of the results found was missing. The discussion is not a summary of the results. You should compare the results obtained with other results available in the literature and find associations that corroborate the data found.

Answer: Thank you for your comment and providing good suggestion for our manuscript. We have added the relevant content to the discussion section. Additional comparison of study findings to prior research literature has also been provided in the discussion. (Page21 line330-332 line340-346; Page22 line352-359 line363-373; Page23 line386-389; Page24 line429-435; Page25 line441-442 line446-459)

Minor issues:

Correct minor errors according to the submission guidelines:

1.Include page numbers and line numbers in the manuscript file. Use continuous line numbers (do not restart the numbering on each page).

Answer: Thank you for your comment and providing good suggestion for our manuscript. We have used page numbers and line numbers in the manuscript file.

2.Figure captions are inserted immediately after the first paragraph in which the figure is cited. Figure files are uploaded separately. Use figure label with Arabic numerals, and “Figure” abbreviated to “Fig” (e.g. Fig 1, Fig 2, Fig 3, etc).

Answer: Thank you for your comment and providing good suggestion for our manuscript. The figure captions were inserted immediately after the first paragraph in which the figures were cited, and the figure labels were also modified according to the guidelines. The figure files were also uploaded separately.

3.Materials and methods

Add the website as a reference instead of placing it in the middle of the text (https://vizhub.healthdata.org/gbd-results/).

Answer: Thank you for your comment and providing good suggestion for our manuscript. We have replaced the website with reference 22 in our manuscript (Page5 line141).

4.Discussion: Please find a more recent reference to replace reference number 19.

Answer: Thank you for your comment and providing good suggestion for our manuscript. We have found a more recent reference to replace reference number 19(Page21 line326).

5."which indicated that the PUD burden was significantly heavier in males, potentially due to lower levels of estrogen" - This assumption needs a reference or better argumentation.

Answer: Thank you for your comment and providing good suggestion for our manuscript. We have added the relevant content and corresponding references accordingly(Page21 line340-345).

6."which appeared to be associated with limited access to diagnostic services." - A reference is needed for this assumption.

Answer: Thank you for your comment and providing good suggestion for our manuscript. We have added recent references for the above sentence(Page23 line401).

Reviewer #2:

(Please refer to the line numbers of the revised version of the manuscript)

Q0: A similar study using the same material was published in 2023: "Peptic ulcer disease burden, trends, and inequalities in 204 countries and territories, 1990–2019: a population-based study" ( doi: 10.1177/17562848231210375). Beyond sex (male patients being more vulnerable in terms of incidence, prevalence mortality and DALY) and smoking (a primary risk factor), were there any other notable findings comparing the 2023 study to the current one? Additionally, did the COVID-19 pandemic influence the epidemiological characteristics of peptic ulcer disease between 2020 and 2021? If not, what are the authors' perspectives on this.

Answer: Thank you for your questions. The latest version of the 2021 dataset was released on the official website of the Global Burden of Disease (GBD) public database on May 16, 2024. To investigate the latest epidemiological situation of Peptic Ulcer Disease (PUD), we utilized the most recent data from 2021 for our analysis. Coupled with the literature you referenced, our study presents the following primary innovations:

1.We analyzed the numbers and age-standardized incidence, prevalence, deaths and DALYs rates of peptic ulcer disease in 2021, whereas the literature analyzed the numbers and age-standardized incidence rates in 2019.

2. We analyzed the age-specific numbers and rates of PUD globally in 2021, whereas the literature did not include related research.

3.We analyzed the trends in the all-age numbers and age-standardized incidence, prevalence, mortality and DALYs rates of PUD by sex from 1990 to 2021, whereas the literature did not include related research.

4. We did the Joinpoint regression analysis of the sex-specific ASRs for PUD globally from 1990 to 2021, whereas the literature did not include related research.

5. We analyzed the age-standardized rates of PUD globally and for 21 GBD regions by SDI from 1990 to 2021, and also analyzed the age-standardized rates of PUD in 204 countries globally by SDI in 2021, whereas the literature analyzed the numbers and age-standardized incidence rates by SDI in 2019.

6. We employed the ARIMA model to predict trends in Age-standardized rates of PUD for both sexes globally over the next 29 years (2022-2050), whereas the literature did not include related research.

7. We analyzed the risk factors for PUD based on the GBD database, whereas the literature did not include related research.

Answer: I think that the COVID-19 pandemic may affect the epidemiological characteristics of PUD between 2020 and 2021. The specific impacts can be analyzed from the following aspects:

1.Redistribution of Medical Resources and Delayed Diagnosis

The COVID-19 pandemic led to a global redistribution of medical resources, particularly during the peak of the pandemic, when many hospitals shifted their focus to treating COVID-19 patients. This resulted in delays or interruptions in the diagnosis and treatment of many routine diseases, including peptic ulcers. Many patients may not have sought medical attention in time, leading to delayed diagnosis and treatment, which in turn affected ulcer incidence and mortality.

2.Helicobacter pylori Transmission and Treatment Interventions

Helicobacter pylori is one of the primary causative factors of peptic ulcers. The COVID-19 pandemic strengthened public health measures, including social distancing and hygiene protocols, which may have influenced the transmission rate of H. pylori. Meanwhile, the pandemic caused treatment delays, and patients might have missed routine H. pylori eradication therapy, potentially worsening peptic ulcers.

3.Changes in Stress and Unhealthy Lifestyle Habits

During the COVID-19 pandemic, there was unprecedented psychological and social stress worldwide. Many people faced unemployment, isolation, and health crises, significantly increasing stress levels. Stress is a key trigger for peptic ulcers, and prolonged anxiety and psychological pressure could exacerbate ulcer development. Additionally, lifestyle changes during the pandemic, such as irregular eating habits, lack of exercise, smoking, and alcohol abuse, may have contributed to the onset and progression of peptic ulcers.

4.Impact of Non-Steroidal Anti-Inflammatory Drug (NSAID) Use

During the COVID-19 pandemic, some patients may have misused non-steroidal anti-inflammatory drugs (NSAIDs) for symptoms such as pain or fever, which could have worsened the occurrence of peptic ulcers. Since NSAIDs are known gastrointestinal ulcer risk factors, particularly with long-term use, this medication pattern during the pandemic may have led to an increased incidence of ulcers.

5.Reduction in Public Health Policies and Medical Screening

At the beginning of the pandemic, many countries restricted health screenings and treatments for non-acute diseases, which may have resulted in delayed diagnosis and treatment for peptic ulcer patients, particularly in areas severely affected by the pandemic. These changes may have influenced the statistical data for PUD in 2020 and 2021, especially in terms of early diagnosis and screening.

In conclusion, the COVID-19 pandemic has had multiple impacts on the epidemiological characteristics of many kinds of diseases, including PUD. Although we observed a potential decrease in ulcer incidence in some regions, factors such as strained medical resources, increased psychological stress, unhealthy lifestyle habits, and treatment delays may have posed challenges to the onset and treatment of peptic ulcers. The combined effect of these factors may have led to delayed diagnosis during the pandemic, and even increased the severity of the disease. Therefore, it is essential to maintain a focus on the prevention, diagnosis, and treatment of peptic ulcers in the post-pandemic context.

Q1: The authors noted in the Discussion section that 'the ASRs of PUD remain abnormally high in Taiwan.' However, in the 2023 study mentioned in Q0 stated that 'EAPCs of ASDR were significantly decreasing in ... Taiwan'. What are the authors' opinions on this?

Answer: Thank you for your question. In our study, we found that Taiwan (Province of China) (335.31 (95% UI: 284.75 to 399.55) per 100,000) had the highest age-standardized prevalence rate (ASPR) in 2021, whereas the literature showed that the EAPCs of age-standardized DALYs rates (ASDR) were significantly decreasing in Taiwan (Province of China) from 1990 to 2019. Futhermore, the EAPC (Estimated Annual Percentage Change) is used to measure the trend of changes in a specific disease indicator (such as incidence rate, mortality rate, or disability-adjusted life years) over a defined period. The above analyses are different both in terms of analytical perspectives and time periods.

Q2: The ASRs of incidence, prevalence, deaths and DALYs of PUD decrease more rapidly in males than in females. Please discuss the possible reasons for this phenomenon.

Answer: Thank you for your comment and providing good suggestion for our manuscript. We have added the corresponding content in the discussion section. Smoking and alcohol consumption have been recognized as risk factors for PUD. Previous research has shown that the prevalence of active smoking among men decreased from 60.0% to 47.6%, while the proportion of daily alcohol consumption declined from 19.0% to 13.1% between 1997 and 2020, which could partly explain the aforementioned decreasing trends for males(Page22 line355-359).

Q3: How can the causal relationship between peptic ulcer disease (PUD)-related disability-adjusted life years (DALYs) and cigarette smoking be determined?

Answer: Thank you for your question. The Population Attributable Fraction (PAF) is the proportion of the outcome that would be removed if a risk factor had been reduced to the theoretical minimum risk exposure level (TMREL). The GBD 2021 utilized the comparative risk assessment framework to estimate DALYs and deaths for PUD attributable to specific risk factors.

PAF Calculation:

PAF=P(RR-1)/P(RR-1) + 1

P represents the prevalence of smoking in the population, and RR is the relative risk of smoking associated with PUD. The GBD attribution analysis model is used to estimate the contribution of smoking to PUD-related DALYs across different populations.

Q4: In "Global and regional burden of PUD", the authors mentioned: "The Taiwan (Province of China) (335.31 (95% UI: 284.75 to 399.55) per 100,000) had the highest ASPR." Though this annotation '(Province of China)' was shown in IHME forms, Taiwan is not globally recognized as a province of China; if it were, China would not pursue its annexation. For instance, does China plan to annex Hebei or Guangdong (both Province of China) in the future? Please remove the annotation '(Province of China)' to avoid potential political issues in scientific research.

Answer: Thank you for your comment and providing good suggestion for our manuscript. We have removed the ann

---

## [Decision Letter · Decision Letter 1]

PONE-D-24-46593R1Global burden and risk factors of peptic ulcer disease between 1990 and 2021: an analysis from the Global Burden of Disease Study 2021PLOS ONE

Dear Dr. Ma,

Thank you for submitting your manuscript to PLOS ONE. After careful consideration, we feel that it has merit but does not fully meet PLOS ONE’s publication criteria as it currently stands. Therefore, we invite you to submit a revised version of the manuscript that addresses the points raised during the review process.

We look forward to receiving your revised manuscript.

Kind regards,

Emmanuel O Adewuyi, BPharm, MPH, PhD

Academic Editor

PLOS ONE

Journal Requirements:

Additional Editor Comments:

One of the reviewers has raised concerns regarding causality and adherence to the correction commitment. The authors need to respond to these comments and make corrections where necessary, or provide a rebuttal as appropriate. Additionally, the authors should carefully review their work sentence by sentence and fact-check every statement.

Reviewers' comments:

Reviewer's Responses to Questions

**Comments to the Author**

1. If the authors have adequately addressed your comments raised in a previous round of review and you feel that this manuscript is now acceptable for publication, you may indicate that here to bypass the “Comments to the Author” section, enter your conflict of interest statement in the “Confidential to Editor” section, and submit your "Accept" recommendation.

Reviewer #1: All comments have been addressed

Reviewer #2: All comments have been addressed

Reviewer #3: All comments have been addressed

2. Is the manuscript technically sound, and do the data support the conclusions?

Reviewer #1: Yes

Reviewer #2: Partly

Reviewer #3: Yes

3. Has the statistical analysis been performed appropriately and rigorously? 

Reviewer #1: Yes

Reviewer #2: I Don't Know

Reviewer #3: Yes

4. Have the authors made all data underlying the findings in their manuscript fully available?

Reviewer #1: Yes

Reviewer #2: Yes

Reviewer #3: Yes

5. Is the manuscript presented in an intelligible fashion and written in standard English?

Reviewer #1: Yes

Reviewer #2: Yes

Reviewer #3: Yes

6. Review Comments to the Author

Reviewer #1: (No Response)

Reviewer #2: Thank you for your response. I have the following comments:

Q1. In response to Q3 from the previous review — "How can the causal relationship between peptic ulcer disease (PUD)-related disability-adjusted life years (DALYs) and cigarette smoking be determined?" — the authors referred to the population attributable fraction (PAF) formula for explanation. However, it is important to note that a "risk factor" or the use of "PAF" does not inherently establish a causal relationship between the factor and the disease.

Q2. In the "Results" and "Discussion" sections (pages 23 and 33, respectively), the manuscript still contains the term "Taiwan (Province of China)," despite the authors' prior agreement in Q4 to remove the "Province of China" annotation. This inconsistency raises concerns about the authors' integrity and adherence to revision commitments. Given this, I recommend that the manuscript be rejected from publication.

Reviewer #3: (No Response)

7. PLOS authors have the option to publish the peer review history of their article (what does this mean? ). If published, this will include your full peer review and any attached files.

**Do you want your identity to be public for this peer review?** For information about this choice, including consent withdrawal, please see our Privacy Policy .

Reviewer #1: No

Reviewer #2: No

Reviewer #3: No

---

## [Author Response · Author response to Decision Letter 2]

12 Apr 2025

Dear academic editor & reviewers,

Thank you for reviewing our manuscript and for the constructive comments, which greatly helped us to improve the manuscript. We have revised the manuscript in accordance with your comments. And point-by-point responses to the comments were as follows:

Journal Requirements:

Answer: Thank you for your kind reminder regarding the reference list. In response, we have conducted a thorough review of all cited references to ensure their completeness, accuracy, and current validity.

We have made the following adjustments:

1.Replacement of improperly formatted or duplicate references:

(1) Reference [5] was found to be formatted incorrectly (e.g., missing journal issue numbers, inconsistent citation styles). New reference [5]: Shen HY, Li X, Ye BX, Jiang JX. A fundic gland polyp and peptic ulcer in Meckel's diverticulum. Dig Liver Dis. 2022 Oct;54(10):1439-1440. https://doi.org/10.1016/j.dld.2022.02.004 PMID: 35236642→ The change has been marked in the revised manuscript on Page27, line545-547.

(2) We identified two instances of duplicate references in the manuscript (previous references [6] and [9] which both cited the same article by Laucirica I et al., 2023). Therefore reference [9] has been removed, and a more relevant, recent article has been cited in its place: Kavitt RT, Lipowska AM, Anyane-Yeboa A, Gralnek IM. Diagnosis and Treatment of Peptic Ulcer Disease. Am J Med. 2019 Apr;132(4):447-456. https://doi.org/10.1016/j.amjmed.2018.12.009 PMID: 30611829 → The change has been marked in the revised manuscript on Page28, line577-579.

(3) We identified two instances of duplicate references in the manuscript (previous references [8] and [10] which both cited the same article by Hudnall A et al., 2022). Therefore reference [10] has been removed, and a more relevant, recent article has been cited in its place: Markar SR, Vidal-Diez A, Patel K, Maynard W, Tukanova K, Murray A, et al. Comparison of Surgical Intervention and Mortality for Seven Surgical Emergencies in England and the United States. Ann Surg. 2019 Nov;270(5):806-812. https://doi.org/10.1097/SLA.0000000000003518 PMID: 31567504→ The change has been marked in the revised manuscript on Page28, line580-584.

2. Addition of DOI and PMID for All References:

We have revised the entire reference list to include both DOI and PMID (where available) for every cited publication. This enhances the traceability and accessibility of cited sources. All updates have been applied to the reference section in the revised manuscript and are reflected in the tracked changes version → The change has been marked in the revised manuscript on Page27-38.

3. Reference Style and Formatting:

Minor inconsistencies in punctuation, author listing, journal abbreviations, and publication dates have been corrected to meet the journal’s style guidelines → The change has been marked in the revised manuscript on Page27-38.

All reference-related revisions have been clearly noted in the main text and are summarized in the rebuttal letter accompanying the revised manuscript. We believe these updates enhance the academic rigor and transparency of our submission.

Thank you again for your guidance and the opportunity to improve our manuscript.

Sincerely,

Huachong Ma

Reviewer's Responses to Questions

Reviewer#2: Thank you for your response. I have the following comments:

Q1. In response to Q3 from the previous review — "How can the causal relationship between peptic ulcer disease (PUD)-related disability-adjusted life years (DALYs) and cigarette smoking be determined?" — the authors referred to the population attributable fraction (PAF) formula for explanation. However, it is important to note that a "risk factor" or the use of "PAF" does not inherently establish a causal relationship between the factor and the disease.

Answer: Thank you for your comment and providing good suggestion for our manuscript. We appreciate your insightful comment regarding the interpretation of the relationship between cigarette smoking and PUD-related DALYs in our manuscript. We fully agree that using the population attributable fraction (PAF) does not inherently establish a causal relationship between a risk factor (such as cigarette smoking) and an outcome (such as peptic ulcer disease). Rather, the GBD framework applies a rigorous set of criteria to determine which risk–outcome pairs are included in the estimation process, and only those with sufficient evidence of causality are assessed using PAF. Specifically, in the Global Burden of Disease (GBD) Study, risk–outcome relationships are evaluated based on comprehensive reviews of the World Cancer Research Fund (WCRF), International Agency for Research on Cancer (IARC), World Health Organization (WHO), and other authoritative sources, alongside systematic literature reviews and meta-analyses. Only when a causal relationship has been judged to be convincing or probable based on Bradford Hill criteria and other epidemiological standards, is the association included in the comparative risk assessment (CRA) and modeled using the PAF formula. In our manuscript, we used PAF values from the GBD 2021 study, which reflect the estimated proportion of PUD burden attributable to smoking, under the assumption of a causal relationship already established through prior rigorous evidence synthesis. We have clarified this point in the revised manuscript (see on Page25, line442-447), adding the following sentence:

“It is important to note that the population attributable fraction (PAF) reflects an estimated burden assuming a causal relationship, which in the context of the GBD study is based on prior evaluation of causal criteria. Therefore, PAF use in this study does not itself establish causality, but quantifies the impact of a risk factor where causal inference has already been determined by the GBD methodology.”

We hope this clarification adequately addresses the concern.

Sincerely,

Huachong Ma

On behalf of the author team

Q2. In the "Results" and "Discussion" sections (pages 23 and 33, respectively), the manuscript still contains the term "Taiwan (Province of China)," despite the authors' prior agreement in Q4 to remove the "Province of China" annotation. This inconsistency raises concerns about the authors' integrity and adherence to revision commitments. Given this, I recommend that the manuscript be rejected from publication.

Answer: Thank you very much for your careful review and valuable comments. We would like to respectfully clarify that the issue mentioned has in fact been addressed in our revised manuscript. Specifically, the relevant revisions were made in accordance with your prior suggestion. The corrected wording appears in the revised manuscript on page 18, line 259:

page 18, line 267-268:

page 23, line 395:

It is possible that the modification was inadvertently overlooked during the re-check, perhaps due to the formatting or its position in the paragraph. We sincerely apologize for any confusion this may have caused. We greatly appreciate your thorough review and your guidance, which have been instrumental in improving the quality and clarity of our manuscript.

Sincerely,

Huachong Ma

---

## [Editor Report · Decision Letter 2]

PONE-D-24-46593R2Global burden and risk factors of peptic ulcer disease between 1990 and 2021: an analysis from the Global Burden of Disease Study 2021PLOS ONE

Dear Dr. Ma,

Thank you for submitting your manuscript to PLOS ONE. After careful consideration, we feel that it has merit but does not fully meet PLOS ONE’s publication criteria as it currently stands. Therefore, we invite you to submit a revised version of the manuscript that addresses the points raised during the review process.

**ACADEMIC EDITOR: **

1. Incomplete Correction Regarding “Taiwan (Province of China)”

Despite the authors’ indication that all instances of 'Province of China' were removed from references to Taiwan, at least one instance remains in the main manuscript. We appreciate that such an error may have been unintentional. However, given this is the second or third time the issue is recurring, the sensitivity of the terminology and prior revision instructions, it is essential that all instances are carefully corrected in the next revision. Please comprehensively review the entire manuscript, including main text, figure captions, tables, and supplementary materials, to ensure full removal of ‘Province of China’ references, consistently referring to the location simply as ‘Taiwan’ as recommended by the reviewer.

2. Methodological Concern: ARIMA Model Projections

Authors use the ARIMA model to project the future burden of peptic ulcer disease up to 2050. However, some projections—notably age-standardised mortality rates (ASMR) and disability-adjusted life-years (ASDR) for females—predict negative rates by 2050. Negative incidence or mortality rates are biologically implausible. Authors need to revise the ARIMA modelling approach to either constrain the lower limit of projections to zero or explain the biological and methodological limitations of ARIMA predictions in the Discussion section and avoid interpreting negative projections literally.

3. Language and Grammar

Although the manuscript is generally understandable, several grammatical errors remain, including but not limited to: typographical errors, incorrect article usage, and redundant wording. A full, careful language edit (sentence-by-sentence) is necessary to ensure clarity and professional presentation. Authors may seek assistance from a professional editing service.

We look forward to receiving your revised manuscript.

Kind regards,

Emmanuel O Adewuyi, BPharm, MPH, PhD

Academic Editor

PLOS ONE
---

## [Author Response · Author response to Decision Letter 3]

8 May 2025

Dear academic editor,

Thank you for reviewing our manuscript and for the constructive comments, which greatly helped us to improve the manuscript. We have revised the manuscript in accordance with your comments. And point-by-point responses to the comments were as follows:

1. Incomplete Correction Regarding “Taiwan (Province of China)”

Despite the authors’ indication that all instances of 'Province of China' were removed from references to Taiwan, at least one instance remains in the main manuscript. We appreciate that such an error may have been unintentional. However, given this is the second or third time the issue is recurring, the sensitivity of the terminology and prior revision instructions, it is essential that all instances are carefully corrected in the next revision. Please comprehensively review the entire manuscript, including main text, figure captions, tables, and supplementary materials, to ensure full removal of ‘Province of China’ references, consistently referring to the location simply as ‘Taiwan’ as recommended by the reviewer.

Answer: We sincerely thank you for pointing out the remaining reference to “Province of China” in relation to Taiwan. We apologize for this oversight and fully acknowledge the importance and sensitivity of this terminology.

In the revised manuscript with track changes, we have conducted a comprehensive review of the entire manuscript, including the main text, figures, figure captions, tables, and supplementary materials. We have carefully removed all instances of “Province of China” and have consistently used “Taiwan” throughout the document, in accordance with the reviewer’s recommendation and the journal’s guidance.

The reference to "Province of China" has been removed both in the main text and in Figure 5, as shown below:

page 19, line 258:

page 20, line 272-273:

page 25, line 406:

Figure 5:

We greatly appreciate your thorough review and your guidance, which have been instrumental in improving the quality and clarity of our manuscript.

Sincerely,

Huachong Ma

2. Methodological Concern: ARIMA Model Projections

Authors use the ARIMA model to project the future burden of peptic ulcer disease up to 2050. However, some projections—notably age-standardised mortality rates (ASMR) and disability-adjusted life-years (ASDR) for females—predict negative rates by 2050. Negative incidence or mortality rates are biologically implausible. Authors need to revise the ARIMA modelling approach to either constrain the lower limit of projections to zero or explain the biological and methodological limitations of ARIMA predictions in the Discussion section and avoid interpreting negative projections literally.

Answer: We appreciate your insightful observation regarding the ARIMA-based projections, specifically the biologically implausible negative values in the age-standardised mortality rate (ASMR) and disability-adjusted life years (ASDR) for females by 2050. We fully agree that negative incidence or mortality rates are not biologically plausible. The ARIMA model, while widely used for time-series forecasting, is purely statistical in nature and does not incorporate biological constraints. As such, it may yield unrealistic values—particularly when the input data demonstrate long-term declining trends or when projections are extended far into the future, as in our case. To address the concern regarding the appearance of implausible (e.g., negative values in the age-standardised rates) in the long-term forecast, we have revised our analysis by shortening the ARIMA model-based prediction period to 2022–2040. This adjustment ensures the stability and interpretability of the results while remaining aligned with the available historical trend data. The relevant S1 file, figure (figure6) and descriptions (page 3, line 80; page 5, line 139; page 7, line 206; page 21, line 277-294; page 25, line 415-416; page 39, line 888-897; page 40, line 909) have been modified and uploaded accordingly.

We thank you again for pointing out this important issue and believe these changes substantially improve the rigor and clarity of our analysis.

Sincerely,

Huachong Ma

3. Language and Grammar

Although the manuscript is generally understandable, several grammatical errors remain, including but not limited to: typographical errors, incorrect article usage, and redundant wording. A full, careful language edit (sentence-by-sentence) is necessary to ensure clarity and professional presentation. Authors may seek assistance from a professional editing service.

Answer: Thank you for your valuable feedback regarding the language quality of our manuscript. In response, we have carefully revised the entire manuscript with close attention to grammar, article usage, typographical errors, and overall clarity. A thorough sentence-by-sentence edit has been conducted to enhance the professionalism and readability of the text. We believe the revised version now meets the journal’s language standards.

Specifically, we made the following changes:

Abstract: Corrected word usage and reworded several sentences to improve clarity and eliminate awkward phrasing (page 2, line 44-48, line 56-58; page 3, line 81-82, line 86-87).

Introduction: Corrected word usage to improve clarity (page 5, line 141).

Methods: Corrected word usage to improve clarity (page 5, line 151-152, line 157).

Results: Enhanced the precision of expressions and corrected word usage to improve clarity and eliminate awkward phrasing (page 8, line 245-248, line 252-253; page 16, line 179-182, line 184-185, line 188-191; page 17, line 201; page 18, line 224, line 237-238).

Discussion and Conclusion: Enhanced the precision of expressions and corrected word usage to improve clarity and eliminate awkward phrasing (page 24, line 390; page 25, line 402-403, line 408, line 420; page 27, line 466).

We hope that the revised version is now acceptable in terms of language quality. Please let us know if further revisions are needed.

Sincerely,

Huachong Ma

Journal Requirements:

Answer: Thank you for your kind reminder regarding the reference list. In response, we have conducted a thorough review of all cited references to ensure their completeness, accuracy, and current validity.

We have made the following adjustments:

1.Replacement of improperly formatted or duplicate references:

(1) Reference [5] was found to be formatted incorrectly (e.g., missing journal issue numbers, inconsistent citation styles). New reference [5]: Shen HY, Li X, Ye BX, Jiang JX. A fundic gland polyp and peptic ulcer in Meckel's diverticulum. Dig Liver Dis. 2022 Oct;54(10):1439-1440. https://doi.org/10.1016/j.dld.2022.02.004 PMID: 35236642→ The change has been marked in the revised manuscript on Page29, line606-608.

(2) We identified two instances of duplicate references in the manuscript (previous references [6] and [9] which both cited the same article by Laucirica I et al., 2023). Therefore reference [9] has been removed, and a more relevant, recent article has been cited in its place: Kavitt RT, Lipowska AM, Anyane-Yeboa A, Gralnek IM. Diagnosis and Treatment of Peptic Ulcer Disease. Am J Med. 2019 Apr;132(4):447-456. https://doi.org/10.1016/j.amjmed.2018.12.009 PMID: 30611829 → The change has been marked in the revised manuscript on Page29, line620-622.

(3) We identified two instances of duplicate references in the manuscript (previous references [8] and [10] which both cited the same article by Hudnall A et al., 2022). Therefore reference [10] has been removed, and a more relevant, recent article has been cited in its place: Markar SR, Vidal-Diez A, Patel K, Maynard W, Tukanova K, Murray A, et al. Comparison of Surgical Intervention and Mortality for Seven Surgical Emergencies in England and the United States. Ann Surg. 2019 Nov;270(5):806-812. https://doi.org/10.1097/SLA.0000000000003518 PMID: 31567504→ The change has been marked in the revised manuscript on Page29, line623-626; Page30, line637.

2. Addition of DOI and PMID for All References:

We have revised the entire reference list to include both DOI and PMID (where available) for every cited publication. This enhances the traceability and accessibility of cited sources. All updates have been applied to the reference section in the revised manuscript and are reflected in the tracked changes version → The change has been marked in the revised manuscript on Page28-39.

3. Reference Style and Formatting:

Minor inconsistencies in punctuation, author listing, journal abbreviations, and publication dates have been corrected to meet the journal’s style guidelines → The change has been marked in the revised manuscript on Page28-39.

All reference-related revisions have been clearly noted in the main text and are summarized in the rebuttal letter accompanying the revised manuscript. We believe these updates enhance the academic rigor and transparency of our submission.

Thank you again for your guidance and the opportunity to improve our manuscript.

Sincerely,

Huachong Ma

---

## [Editor Report · Decision Letter 3]

PONE-D-24-46593R3Global burden and risk factors of peptic ulcer disease between 1990 and 2021: an analysis from the Global Burden of Disease Study 2021PLOS ONE

Dear Dr. Ma,

Thank you for submitting your manuscript to PLOS ONE. After careful consideration, we feel that it has merit but does not fully meet PLOS ONE’s publication criteria as it currently stands. Therefore, we invite you to submit a revised version of the manuscript that addresses the points raised during the review process.

**ACADEMIC EDITOR: ** Thank you for your revised submission. The manuscript is greatly improved, and your responses have addressed the reviewers’ and editor’s concerns. However, a few minor issues remain that require correction before the manuscript can be accepted.

First, peptic ulcer disease (PUD) is described as a ‘prevalent acute abdominal condition’, which could be misleading. PUD is a chronic gastrointestinal condition that may present acutely when complications arise. Please revise this description for accuracy or provide justification for using ‘acute’.

Second, there are minor discrepancies between the structured abstract and the full abstract, including a spelling error (‘identifed’ should be ‘identified’) and some formatting inconsistencies.

Finally, please take this opportunity to review the manuscript sentence by sentence to ensure accuracy, clarity, and consistency throughout.

We look forward to receiving your revised manuscript.

Kind regards,

Emmanuel O Adewuyi, BPharm, MPH, PhD

Academic Editor

PLOS ONE
---

## [Author Response · Author response to Decision Letter 4]

16 May 2025

Dear academic editor,

Thank you for reviewing our manuscript and for the constructive comments, which greatly helped us to improve the manuscript. We have revised the manuscript in accordance with your comments. And point-by-point responses to the comments were as follows:

ACADEMIC EDITOR: Thank you for your revised submission. The manuscript is greatly improved, and your responses have addressed the reviewers’ and editor’s concerns. However, a few minor issues remain that require correction before the manuscript can be accepted.

1. First, peptic ulcer disease (PUD) is described as a ‘prevalent acute abdominal condition’, which could be misleading. PUD is a chronic gastrointestinal condition that may present acutely when complications arise. Please revise this description for accuracy or provide justification for using ‘acute’.

Answer: Thank you for your valuable feedback regarding the description of peptic ulcer disease (PUD). We agree that referring to PUD as a “prevalent acute abdominal condition” may lead to misunderstanding, given its chronic nature. Accordingly, we have revised the manuscript to describe PUD as a chronic gastrointestinal condition that can present acutely when complications occur. This revision clarifies that while PUD itself is chronic, acute presentations often necessitate urgent clinical attention.

The revised sentence now reads:

“Peptic ulcer disease (PUD) is a chronic gastrointestinal disorder that may present acutely due to complications and poses significant clinical and economic challenges.” (page 2, line 42-44)

We believe this change accurately reflects the clinical characteristics of PUD and aligns with current medical understanding.

2. Second, there are minor discrepancies between the structured abstract and the full abstract, including a spelling error (‘identifed’ should be ‘identified’) and some formatting inconsistencies.

Answer: Thank you for pointing out the discrepancies between the structured abstract and the full abstract. We have carefully reviewed both sections to ensure consistency in content, structure, and formatting. Specifically, the typographical error “identifed” has been corrected to “identified” (page 3, line 85), and the expressions were refined and word usage corrected to enhance clarity and eliminate awkward phrasing. (page 3, line 85-86, line 88)

We appreciate your attention to detail and believe these corrections enhance the clarity and presentation of the manuscript.

3.Finally, please take this opportunity to review the manuscript sentence by sentence to ensure accuracy, clarity, and consistency throughout.

Answer: We sincerely appreciate your recommendation. In response, we have conducted a thorough, sentence-by-sentence revision of the entire manuscript to enhance its linguistic accuracy, clarity, and consistency. Specifically, we:

• Refined grammatical structures and improved word choices throughout the revised manuscript.

• Eliminated redundant or awkward phrasing to ensure smooth and professional expression.

• Corrected all identified typographical and formatting inconsistencies, including in the structured and unstructured abstracts.

• Ensured consistency in terminology and abbreviations.

We hope these efforts meet the expectations of the editorial team and enhance the overall readability and quality of the manuscript.

---

## [Editor Report · Decision Letter 4]

Global burden and risk factors of  peptic ulcer disease between 1990 and 2021: an analysis from the Global Burden of Disease Study 2021

PONE-D-24-46593R4

Dear Dr. Ma,

We’re pleased to inform you that your manuscript has been judged scientifically suitable for publication and will be formally accepted for publication once it meets all outstanding technical requirements.

Kind regards,

Emmanuel O Adewuyi, BPharm, MPH, PhD

Academic Editor

PLOS ONE
---

## [Editor Report · Acceptance letter]

PONE-D-24-46593R4

PLOS ONE

Dear Dr. Ma,

I'm pleased to inform you that your manuscript has been deemed suitable for publication in PLOS ONE. Congratulations! Your manuscript is now being handed over to our production team.

Kind regards,

on behalf of

Dr. Emmanuel O Adewuyi

Academic Editor

PLOS ONE